# When VMP Meets CEP: An Algorithmic Equivalence Under Mild Conditions

**Siyuan Li**    *lisiyuan@zju.edu.cn*
*College of Information Science and Electronic Engineering*
*Zhejiang University*

**Shikai Fang**    *fsk@zju.edu.cn*
*College of Information Science and Electronic Engineering*
*Zhejiang University*

**Lei Cheng**\*    *lei\_cheng@zju.edu.cn*
*College of Information Science and Electronic Engineering*
*Zhejiang University*

**Yik-Chung Wu**    *ycwu@eee.hku.hk*
*Department of Electrical and Electronic Engineering*
*The University of Hong Kong*

**Sergios Theodoridis**    *stheodor@di.uoa.gr*
*HERON - Hellenic Robotics Center of Excellence, Athena R.C., Greece*
*National and Kapodistrian University of Athens, Greece*

**Reviewed on OpenReview:** *https://openreview.net/forum?id=QdO4VrnNfb*

## Abstract

Approximate Bayesian inference (ABI) methods have become indispensable tools in modern machine learning and statistics for approximating intractable posterior distributions. Despite extensive studies, the theoretical connections among different ABI methods have remained relatively unexplored. This paper establishes an algorithmic equivalence between two widely employed ABI techniques, namely variational message passing (VMP) and conditional expectation propagation (CEP). Through rigorous mathematical analysis, we demonstrate that these two approaches, despite originating from different perspectives (variational inference and expectation propagation, respectively), yield the same update equations under mild conditions, from both optimization and graphical model viewpoints. As a direct consequence, we establish a convergence guarantee for CEP and show that VMP-derived algorithms can inherit streaming variants without additional derivation effort. To validate our theoretical findings, we apply both VMP and CEP to Bayesian tensor decomposition and verify that they produce identical updates, demonstrating how the equivalence provides a principled route to a streaming variant.

## 1 Introduction

Approximating difficult-to-compute posterior distributions is one of the most fundamental challenges in modern machine learning and statistics. To address this challenge, approximate Bayesian inference (ABI) has made significant progress over the years (Blei et al., 2017; Zhang et al., 2019; Theodoridis, 2025; Cheng et al., 2022b; Murphy, 2022), showcasing remarkable performance. These methods have found extensive applications in diverse domains, such as bioinformatics (Daunizeau et al., 2014; Grønbech et al., 2020),

---

\*Corresponding author

computer vision (Chan & Vasconcelos, 2009; Soh & Cho, 2022; Fan et al., 2022), and speech recognition (Cohen & Smith, 2010; Xue et al., 2021). Variational inference (VI) (Jordan et al., 1999; Wainwright & Jordan, 2008) and expectation propagation (EP) (Minka & Picard, 2001), along with their modern variants (Zhang et al., 2019; Broderick et al., 2013; Li et al., 2015; Wang & Zhe, 2020; Vehtari et al., 2020), are two prominent classes of ABI methods widely used in practice.

The fundamental principle of VI involves formulating a family of distributions and subsequently finding the member within that family that best approximates the target distribution (Blei et al., 2017; Bishop, 2006; Theodoridis, 2025). The closeness between distributions is typically measured using the Kullback-Leibler (KL) divergence. In the context of the mean-field VI, the variables are assumed to be mutually independent and governed by their respective distributions. By decomposing the model evidence, VI transforms its objective into optimizing the evidence lower bound (ELBO). When analytical expectations can be derived, VI demonstrates favorable accuracy and speed. It also guarantees convergence to a local optimum (Beal, 2003). The streaming version of VI has also been developed to handle the streaming data case (Broderick et al., 2013), as illustrated in Fig. 1.

When the distributions of variables are restricted to the exponential family (Brown, 1986) and possess conjugate properties, mean-field VI can be implemented based on the convenient and efficient message-passing mechanism. The resulting algorithm is known as the variational message passing (VMP) (Winn et al., 2005), as illustrated in Fig. 1. VMP operates by sending messages between nodes in the network and updating posterior beliefs through local operations performed at each node. By introducing additional variational parameters or utilizing approximation methods, VMP can be extended to models containing non-conjugate distributions (Winn et al., 2005; Wang & Blei, 2013). It also guarantees convergence and enables efficient evaluation of the model evidence (Winn et al., 2005).

EP is a generalized message-passing algorithm employed on factor graphs (Minka, 2013), which unifies and extends the concepts of assumed density filtering (ADF) (Maybeck, 1982) and loopy belief propagation (Frey & MacKay, 1997). The ADF can be viewed as a streaming or online version of EP, as illustrated in Fig. 1. In EP, we construct an approximation of the posterior by iteratively performing simple local computations which refine the factor that represents how each data point contributes to the posterior distribution. Notably, EP differs from VI in terms of the direction of the KL divergence. In various tasks, such as the clutter problem[1] and mixture weight estimation, EP has shown superior performance compared to VI (Zhou et al., 2023). Additionally, the local computations make EP amenable to parallelized and distributed computation, rendering it well-suited for addressing large-scale problems (Li et al., 2015; Hasenclever et al., 2017; Vehtari et al., 2020). However, applying EP encounters a critical challenge when dealing with models that have complex likelihoods, as the moment matching that is involved in the factor update procedure can become intractable. Additionally, convergence is not guaranteed as message passing updates are local (Vehtari et al., 2020).

To address the computation barrier in EP, recent advances have introduced alternative approaches for moment computation in EP, such as the Monte Carlo simulations (Li et al., 2018) and the Laplace approximation (Smola et al., 2004). Unfortunately, these approximations often suffer from inefficiency and high computational costs, thereby diminishing the appeal of EP as a fast approximation method. Conditional expectation propagation (CEP) (Wang & Zhe, 2020) has recently emerged as a promising alternative, offering an efficient variant of EP. Instead of directly calculating the moments of the complete distribution, CEP first seeks the tractable and analytical conditional moments and then computes their expectations with respect to the approximate posterior of the remaining variables, as illustrated in Fig. 1. Like EP, CEP's local update nature makes it well-suited for large-scale datasets, but a convergence guarantee remains an open question.

Since VMP and CEP are developed from different perspectives (VI and EP, respectively) and have distinct theoretical roots (different directions of the KL divergence), theoretical connections between them remain unexplored. To the best of our knowledge, the most related line of work focuses on unifying VI and EP through the perspective of divergence minimization. Notable examples include Power EP (Minka, 2004) and Rényi Divergence Variational Inference (Li & Turner, 2016), which unify VI and EP by utilizing generalized

---

[1]The clutter problem is about estimating the mean of a signal distribution when the data is mixed with noise from a known clutter distribution (Minka, 2013).

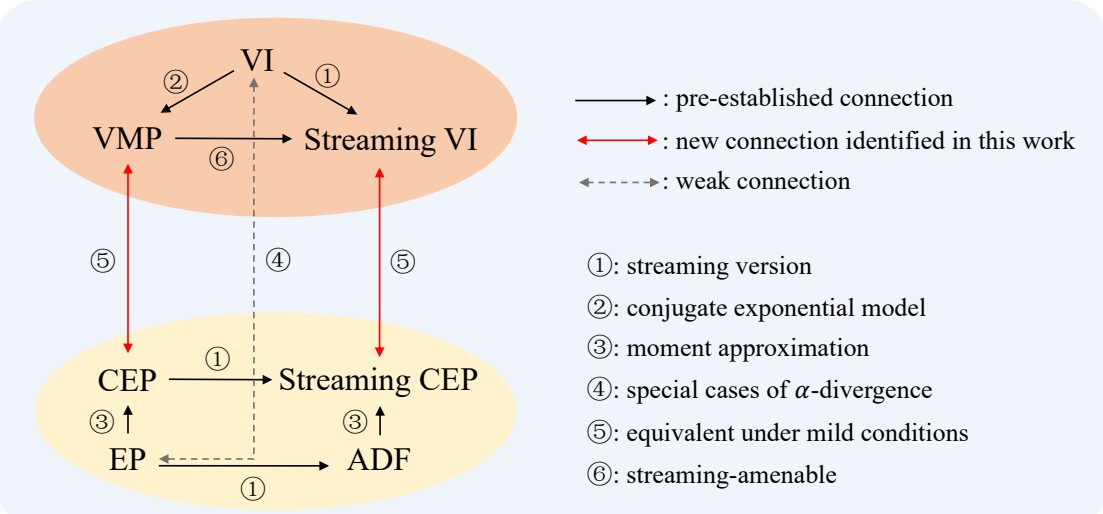

Figure 1: Connections among different approximate Bayesian inference (ABI) methods. Black arrows indicate previously known connections, while red arrows highlight the new algorithmic links established in this paper. VI denotes variational inference, VMP stands for variational message passing, EP refers to expectation propagation, CEP is conditional expectation propagation, and ADF represents assumed density filtering. Streaming CEP denotes the streaming variant of CEP, obtained by applying the surrogate distribution and delta approximations to ADF.

divergences, such as the $\alpha$-divergence or Rényi divergence. By adjusting the divergence parameter, these frameworks allow for interpolation between the behaviors of VI and EP. However, while these works provide a general perspective that connects the objectives of VI and EP, they inherently treat them as distinct approaches governed by different divergence parameters. Consequently, they do not uncover the intrinsic connections at the algorithmic level. In particular, they do not reveal how the update equations of VMP and CEP can become mathematically equivalent under specific conditions. In a related direction, Bui et al. (2017) showed that Power EP recovers VI in the limiting case of the power parameter, providing further evidence that VI and EP are closely related. However, this result characterizes the limiting behavior of a generalized divergence rather than establishing an explicit algorithmic equivalence between distinct methods. Another related work is the Bayesian learning rule (Khan & Rue, 2023), which unifies different ABI methods through the natural gradient descent. However, it does not consider the EP algorithm and its variants.

This paper aims to establish the algorithmic equivalence between VMP and CEP under mild conditions, as illustrated by the red arrows in Fig. 1. In particular, we demonstrate that these two approaches yield the same update equations, from both optimization and graphical model perspectives. Notably, the established equivalence provides a convergence guarantee for CEP, leveraging the corresponding property enjoyed by VMP. It turns out that the assimilation of the message factors in CEP leads to an increment of ELBO and ensures convergence. Furthermore, the equivalence provides a principled recipe for deriving streaming variants of VMP-based algorithms, connecting to existing streaming and distributed Bayesian methods (Broderick et al., 2013; Masegosa et al., 2016).

To corroborate our theoretical analysis, we present an example that showcases the application of VMP and CEP in the context of Bayesian tensor decomposition, which is a powerful tool in machine learning research and finds applications in various real-world scenarios (Cheng et al., 2022b;a; Fang et al., 2021b;a). In this particular context, we verify that VMP and CEP produce identical updates and demonstrate how the equivalence naturally yields a streaming variant without additional derivation effort.

The remainder of this paper is organized as follows. Section 2 gives a brief review of different ABI methods and provides some useful lemmas. Section 3 contains the main theoretical results and some related extensions. In Section 4, using Bayesian tensor decomposition as an example, different ABI methods are derived and

analyzed to validate our theoretical findings. Finally, Section 5 concludes with an overall discussion and suggestions for future research directions.

## 2 Preliminaries

This section provides a brief review of various ABI methods and presents some useful lemmas. Before delving into the details of each method, we introduce the general problem. Given a set of observations $\mathcal{D} = \{\mathbf{x}_1, \cdots, \mathbf{x}_N\}$ and a probabilistic model described via a set of latent variables $\boldsymbol{\theta}$, the joint distribution can be expressed as

$$p(\boldsymbol{\theta}, \mathcal{D}) = p(\boldsymbol{\theta})p(\mathcal{D}|\boldsymbol{\theta}),$$

where $p(\boldsymbol{\theta})$ represents the respective prior distribution and $p(\mathcal{D}|\boldsymbol{\theta})$ denotes the data likelihood. The goal is to compute the posterior distribution, $p(\boldsymbol{\theta}|\mathcal{D})$, which can be expressed as

$$p(\boldsymbol{\theta}|\mathcal{D}) = \frac{p(\boldsymbol{\theta}, \mathcal{D})}{p(\mathcal{D})} = \frac{p(\boldsymbol{\theta}, \mathcal{D})}{\int p(\boldsymbol{\theta}, \mathcal{D})d\boldsymbol{\theta}},$$

where $p(\mathcal{D})$ denotes the model evidence. For many models of practical interest, it is infeasible to compute the posterior distribution directly due to the analytically intractable integration in the denominator. Therefore, approximation methods are essential in such cases. In this paper, we primarily focus on VI, EP, and their variants.

### 2.1 Variational Inference and Variational Message Passing

#### 2.1.1 Variational Inference (VI)

VI is a technique that approximates the posterior distribution by utilizing a probability distribution with density $q(\boldsymbol{\theta})$ from a tractable family of distributions $\mathcal{Q}$. The aim is to find the best variational approximation, $q^* \in \mathcal{Q}$, by minimizing the KL divergence between $q(\boldsymbol{\theta})$ and the true posterior $p(\boldsymbol{\theta}|\mathcal{D})$ (Cover, 1999), i.e.,

$$q^* = \min_{q \in \mathcal{Q}} \mathrm{KL}(q(\boldsymbol{\theta})\|p(\boldsymbol{\theta}|\mathcal{D})) = \min_{q \in \mathcal{Q}} \int q(\boldsymbol{\theta}) \ln \frac{q(\boldsymbol{\theta})}{p(\boldsymbol{\theta}|\mathcal{D})} d\boldsymbol{\theta}.$$

This transforms the inference task into an optimization problem, where the flexibility of the family $\mathcal{Q}$ controls the complexity of the optimization process. However, the objective function is not directly computable as it requires the model evidence. To overcome this challenge, VI employs a clever decomposition (e.g., Bishop, 2006; Theodoridis, 2025)

$$\ln p(\mathcal{D}) = \mathcal{L}(q) + \mathrm{KL}(q(\boldsymbol{\theta})\|p(\boldsymbol{\theta}|\mathcal{D})),$$

where

$$\mathcal{L}(q) = \int q(\boldsymbol{\theta}) \ln \frac{p(\boldsymbol{\theta}, \mathcal{D})}{q(\boldsymbol{\theta})} d\boldsymbol{\theta} \tag{1}$$

is the evidence lower bound (ELBO). Since the model evidence is a constant with respect to $\boldsymbol{\theta}$ and the KL divergence is non-negative, minimizing the latter is equivalent to maximizing $\mathcal{L}(q)$.

If there are no restrictions on $\mathcal{L}(q)$, the maximum of the ELBO occurs when $q(\boldsymbol{\theta})$ equals $p(\boldsymbol{\theta}|\mathcal{D})$, which, however, is intractable. Consequently, some restrictions on the functional form of $q(\boldsymbol{\theta})$ are required. In the context of the mean-field VI, the variables are assumed to be mutually independent, and each variable is governed by its own distribution. A typical member of the mean-field variational family can be expressed as (e.g., Blei et al., 2017; Theodoridis, 2025)

$$q(\boldsymbol{\theta}) = \prod_{m=1}^{M} q(\boldsymbol{\theta}_m). \tag{2}$$

Here the elements of $\boldsymbol{\theta}$ are partitioned into $M$ disjoint groups as the probability distribution is factorized into $M$ groups, i.e., $\boldsymbol{\theta} = \{\boldsymbol{\theta}_1, \cdots, \boldsymbol{\theta}_M\}$. Then, the ELBO $\mathcal{L}(q)$ is optimized by iteratively updating each group in turn. Specifically, the optimal solution for each factor can be obtained by substituting (2) into (1), which gives

$$
\begin{aligned}
\mathcal{L}(q) &= \int \prod_m q(\boldsymbol{\theta}_m) \left\{ \ln p(\boldsymbol{\theta}, \mathcal{D}) - \sum_m \ln q(\boldsymbol{\theta}_m) \right\} d\boldsymbol{\theta} \\
&= \int q(\boldsymbol{\theta}_m) \mathbb{E}_{q(\boldsymbol{\theta}_{\backslash m})}[\ln p(\boldsymbol{\theta}, \mathcal{D})] d\boldsymbol{\theta}_m - \sum_j \int q(\boldsymbol{\theta}_j) \ln q(\boldsymbol{\theta}_j) d\boldsymbol{\theta}_j \\
&= \int q(\boldsymbol{\theta}_m) \mathbb{E}_{q(\boldsymbol{\theta}_{\backslash m})}[\ln p(\boldsymbol{\theta}, \mathcal{D})] d\boldsymbol{\theta}_m - \int q(\boldsymbol{\theta}_m) \ln q(\boldsymbol{\theta}_m) d\boldsymbol{\theta}_m + \text{const1} \\
&= -\text{KL}\left(q(\boldsymbol{\theta}_m) \| \tilde{q}(\boldsymbol{\theta}_m)\right) + \text{const1},
\end{aligned}
$$

where $\boldsymbol{\theta}_{\backslash m}$ represents the set of variables excluding the $m$th group and

$$
\ln \tilde{q}(\boldsymbol{\theta}_m) = \mathbb{E}_{q(\boldsymbol{\theta}_{\backslash m})}[\ln p(\boldsymbol{\theta}, \mathcal{D})] + \text{const2}.
$$

It can be seen that $\mathcal{L}(q)$ is optimized when the KL divergence equals zero, which results in

$$
\begin{aligned}
\ln q^*(\boldsymbol{\theta}_m) &= \mathbb{E}_{q(\boldsymbol{\theta}_{\backslash m})}[\ln p(\boldsymbol{\theta}, \mathcal{D})] + \text{const2} \\
&= \mathbb{E}_{q(\boldsymbol{\theta}_{\backslash m})}[\ln p(\boldsymbol{\theta}_m | \boldsymbol{\theta}_{\backslash m}, \mathcal{D})] + \text{const3}.
\end{aligned} \tag{3}
$$

After taking the exponential of both sides and normalizing, we obtain

$$
q^*(\boldsymbol{\theta}_m) = \frac{\exp(\mathbb{E}_{q(\boldsymbol{\theta}_{\backslash m})}[\ln p(\boldsymbol{\theta}_m | \boldsymbol{\theta}_{\backslash m}, \mathcal{D})])}{\int \exp(\mathbb{E}_{q(\boldsymbol{\theta}_{\backslash m})}[\ln p(\boldsymbol{\theta}_m | \boldsymbol{\theta}_{\backslash m}, \mathcal{D})]) d\boldsymbol{\theta}_m}, \forall m. \tag{4}
$$

Although the set of equations in (4) provides consistency conditions for maximizing the lower bound, they do not represent an explicit solution. This is because the optimum for each variable group $\boldsymbol{\theta}_m$ depends on the distributions of other groups, $\boldsymbol{\theta}_{\backslash m}$. Therefore, when applying VI, we typically seek a solution by first initializing all of the factors $q(\boldsymbol{\theta}_m)$ appropriately and then iteratively updating each factor, replacing the other variable groups with their current estimates. Convergence is guaranteed because the ELBO is convex with respect to each of the groups (e.g., Bishop, 2006). It is also worth noting that if the goal is to minimize the reverse KL divergence, i.e., $\text{KL}(p\|q)$, a closed-form solution for each variable group can also be derived; see Appendix B for details.

### 2.1.2 Variational Message Passing (VMP)

VMP is an implementation of the mean-field variational inference method tailored to conjugate-exponential models (Winn et al., 2005). It performs inference by passing local messages in a Bayesian network, allowing structured and modular updates of the variational distributions.

Before introducing the details of VMP, we briefly review the basic concepts of probabilistic graphical models. In a Bayesian network, the joint distribution over a set of variables is represented using a directed acyclic graph (DAG), where each node corresponds to a random variable, and directed edges represent conditional dependencies. The joint distribution factorizes as a product of conditional distributions, one for each node given its parents in the graph.

Fig. 2(a) illustrates a typical Bayesian network structure. The latent variables $\boldsymbol{\theta} = \{\boldsymbol{\theta}_1, \ldots, \boldsymbol{\theta}_M\}$ represent different components of the model parameters, each associated with its own set of parents, denoted by $\text{pa}_m$. These parents may include hyperparameters or other latent variables. The observed data $\{\mathbf{x}_i\}_{i=1}^N$ are conditionally dependent on all or a subset of the latent variables.

In VMP, the probabilistic model is a conjugate-exponential model. Specifically, the distributions of variables/nodes, conditioned on their parents, are drawn from the exponential family and are conjugate with

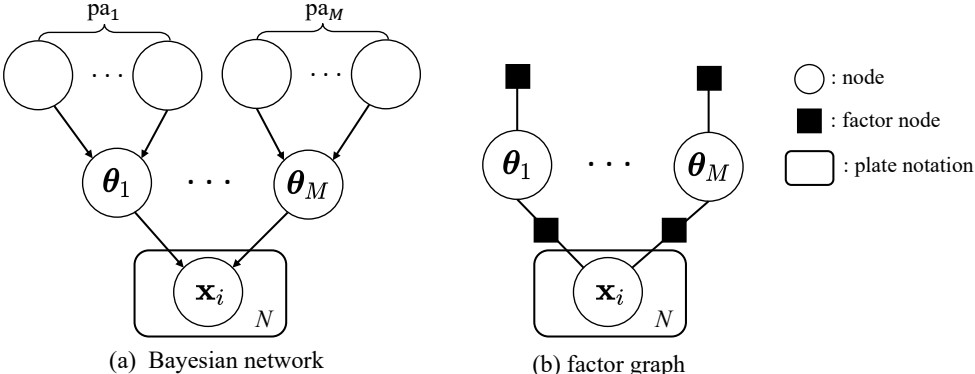

Figure 2: Two graphical model representations: (a) a Bayesian network used in variational message passing. Each latent variable $\boldsymbol{\theta}_m$ has a parent set $\mathrm{pa}_m$, and the observed variables $\{\mathbf{x}_i\}_{i=1}^N$ are conditionally independent given the latent variables; (b) a factor graph used in expectation propagation. Circles denote latent variables $\{\boldsymbol{\theta}_1, \ldots, \boldsymbol{\theta}_M\}$ and observed variable $\mathbf{x}_i$, while squares represent factors. The plate indicates $N$ repeated observations.

respect to the distributions over these parent variables. As a result, each complete conditional is also in the exponential family (e.g., Blei et al., 2017), i.e.,

$$p(\boldsymbol{\theta}_m|\boldsymbol{\theta}_{\backslash m}, \mathcal{D}) = h(\boldsymbol{\theta}_m) \exp\left\{\boldsymbol{\eta}_m(\boldsymbol{\theta}_{\backslash m}, \mathcal{D})^T \boldsymbol{\phi}(\boldsymbol{\theta}_m) - Z_m(\boldsymbol{\eta}_m(\boldsymbol{\theta}_{\backslash m}, \mathcal{D}))\right\}, \tag{5}$$

where $h(\boldsymbol{\theta}_m)$ is a scaling constant; $\boldsymbol{\phi}(\boldsymbol{\theta}_m)$ is the vector of sufficient statistics; $\boldsymbol{\eta}_m$ are the natural parameters; and $Z_m(\cdot)$ is the log partition function. The subscript $m$ indicates that these quantities may vary across different nodes. For simplicity, here, we consider each group $\boldsymbol{\theta}_m$ to contain a single variable.

In the conjugate-exponential model, the update for node $m$ in the mean-field VI becomes significantly simplified. By substituting (5) into (4), the update can be expressed as (e.g., Blei et al., 2017)

$$q^*(\boldsymbol{\theta}_m) \propto \exp(\mathbb{E}_{q(\boldsymbol{\theta}_{\backslash m})}[\ln p(\boldsymbol{\theta}_m|\boldsymbol{\theta}_{\backslash m}, \mathcal{D})]) \tag{6}$$
$$= \exp\left\{\ln h(\boldsymbol{\theta}_m) + \mathbb{E}_{q(\boldsymbol{\theta}_{\backslash m})}[\boldsymbol{\eta}_m(\boldsymbol{\theta}_{\backslash m}, \mathcal{D})]^T \boldsymbol{\phi}(\boldsymbol{\theta}_m) - \mathbb{E}_{q(\boldsymbol{\theta}_{\backslash m})}[Z_m(\boldsymbol{\eta}_m(\boldsymbol{\theta}_{\backslash m}, \mathcal{D}))]\right\}$$
$$\propto h(\boldsymbol{\theta}_m) \exp\left\{\mathbb{E}_{q(\boldsymbol{\theta}_{\backslash m})}[\boldsymbol{\eta}_m(\boldsymbol{\theta}_{\backslash m}, \mathcal{D})]^T \boldsymbol{\phi}(\boldsymbol{\theta}_m)\right\}.$$

This reveals that the optimal variational distribution for a node has the same functional form as the corresponding prior distribution, indicating that we only need to update the parameters of the corresponding distribution. Furthermore, the updates for each one of the nodes can be implemented locally using the expected values (messages) from the other nodes. VMP involves the exchange of messages between nodes in the network and iteratively updating the posterior distribution until the convergence is reached (Li et al., 2024). The detailed algorithm for VMP is summarized in Appendix A.

## 2.2 Expectation Propagation and Conditional Expectation Propagation

### 2.2.1 Expectation Propagation (EP)

EP is a message-passing algorithm that generalizes ADF and loopy belief propagation. Similar to VI, EP also approximates the posterior distribution by minimizing the KL divergence, but in the reverse direction. EP assumes that the joint distribution of the probabilistic model can be written in a factorized form:

$$p(\boldsymbol{\theta}, \mathcal{D}) = \prod_i f_i(\boldsymbol{\theta}),$$

where each $f_i(\boldsymbol{\theta})$ is called a factor and depends only on a subset of the variables. The range of index $i$ is determined by how the variables are grouped or partitioned in the model.

This factorization can be represented using a *factor graph*, which is a bipartite graph consisting of variable nodes and factor nodes. Variable nodes correspond to individual components of $\boldsymbol{\theta}$, while factor nodes correspond to the functions $f_i$. An edge is placed between a variable node and a factor node if the variable appears in that factor. This structure clearly shows how different variables are involved in different parts of the model.

In the case of independently and identically distributed (i.i.d.) observations, each factor $f_i(\boldsymbol{\theta})$ may correspond to the likelihood term $p(\mathbf{x}_i|\boldsymbol{\theta})$ for the $i$th data point, while $f_0(\boldsymbol{\theta})$ represents the prior distribution $p(\boldsymbol{\theta})$. Then, the joint distribution is written as

$$p(\boldsymbol{\theta}, \mathcal{D}) = p(\boldsymbol{\theta}) \prod_{i=1}^{N} p(\mathbf{x}_i|\boldsymbol{\theta}) = \prod_{i=0}^{N} f_i(\boldsymbol{\theta}). \tag{7}$$

Figure 2(b) illustrates a typical factor graph used in EP, where messages are passed between variable nodes and factor nodes based on the graph structure.

We are interested in evaluating the posterior distribution

$$p(\boldsymbol{\theta}|\mathcal{D}) = \frac{p(\boldsymbol{\theta}, \mathcal{D})}{p(\mathcal{D})} = \frac{1}{Z} \prod_i f_i(\boldsymbol{\theta}),$$

where $Z = p(\mathcal{D})$ is the normalization constant, which can be calculated as

$$Z = \int \prod_i f_i(\boldsymbol{\theta}) d\boldsymbol{\theta}.$$

EP approximates the posterior by a product of factors, given by (Minka, 2013)

$$q(\boldsymbol{\theta}) = \frac{1}{\tilde{Z}} \prod_i \tilde{f}_i(\boldsymbol{\theta}),$$

where $\tilde{f}_i$ is an approximation of $f_i$ that belongs to the exponential family, and $\tilde{Z}$ is the associated normalization constant. Ideally, the determination of the involved factors $\{\tilde{f}_i(\boldsymbol{\theta})\}_{i=1}^{N}$ involves the minimization of the KL divergence from $p(\boldsymbol{\theta}|\mathcal{D})$ to $q(\boldsymbol{\theta})$, given by

$$\mathrm{KL}(p\|q) = \mathrm{KL}\left(\frac{1}{Z} \prod_i f_i(\boldsymbol{\theta}) \| \frac{1}{\tilde{Z}} \prod_i \tilde{f}_i(\boldsymbol{\theta})\right).$$

However, this minimization is typically intractable due to the need to compute expectations with respect to the true distribution.

EP approximates the posterior by iteratively refining each factor while accounting for the influence of the others. Specifically, EP follows four steps in each iteration (Minka, 2013). First, select a factor $\tilde{f}_i$ for updating and remove it from the approximation $q(\boldsymbol{\theta})$ to produce the *calibrating* distribution $q^{\backslash i}(\boldsymbol{\theta})$, defined as $q^{\backslash i}(\boldsymbol{\theta}) = q(\boldsymbol{\theta})/\tilde{f}_i(\boldsymbol{\theta})$. Note that $q^{\backslash i}$ can also be derived from the product of factors $i \neq j$, but in practice, the division is more convenient. Second, the calibrating distribution is combined with the factor $f_i(\boldsymbol{\theta})$ to obtain the *tilted* distribution

$$\hat{p}_i(\boldsymbol{\theta}) = \frac{1}{Z_i} f_i(\boldsymbol{\theta}) q^{\backslash i}(\boldsymbol{\theta}), \tag{8}$$

where $Z_i$ is the associated normalization constant. Third, we obtain an approximation $q^{\natural}(\boldsymbol{\theta})$ of $\hat{p}_i(\boldsymbol{\theta})$ by minimizing the KL divergence between $\hat{p}_i(\boldsymbol{\theta})$ and $q^{\natural}(\boldsymbol{\theta})$. If $q^{\natural}(\boldsymbol{\theta})$ belongs to the exponential family, as it is often the case, the minimum can be obtained by moment matching (Maybeck, 1982), i.e.,

$$\mathbb{E}_{q^{\natural}(\boldsymbol{\theta})}[\boldsymbol{\phi}(\boldsymbol{\theta})] = \mathbb{E}_{\hat{p}_i(\boldsymbol{\theta})}[\boldsymbol{\phi}(\boldsymbol{\theta})], \tag{9}$$

where $\boldsymbol{\phi}(\boldsymbol{\theta})$ is the sufficient statistics of $q^\natural(\boldsymbol{\theta})$. A more detailed discussion of moment matching is provided in Appendix B. Note that the natural parameter of $q^\natural(\boldsymbol{\theta})$ is implicitly specified in the moment matching process. For example, if $q^\natural(\boldsymbol{\theta})$ is a Gaussian distribution $\mathcal{N}(\boldsymbol{\theta}|\boldsymbol{\mu}, \boldsymbol{\Sigma})$ then we can minimize the KL divergence by setting the mean $\boldsymbol{\mu}$ equal to the mean of $\hat{p}_i(\boldsymbol{\theta})$ and the covariance $\boldsymbol{\Sigma}$ equal to the covariance of $\hat{p}_i(\boldsymbol{\theta})$. Finally, the factor $\tilde{f}_i$ is updated via $\tilde{f}_i(\boldsymbol{\theta}) \propto q^\natural(\boldsymbol{\theta})/q^{\backslash i}(\boldsymbol{\theta})$.

The rationale behind this update is to ensure that the approximate factor contributes to the posterior in a manner similar to the corresponding data likelihood. Due to the local refinement[2], the factors can be efficiently calculated in a distributed manner. However, convergence is not guaranteed in general.

### 2.2.2 Conditional Expectation Propagation (CEP)

While EP is known for its favorable accuracy and speed on diverse tasks, a significant challenge in its application arises from the computational intractability of the expectations $\mathbb{E}_{\hat{p}_i(\boldsymbol{\theta})}[\boldsymbol{\phi}(\boldsymbol{\theta})]$ in (9) for models with complex data likelihood. To overcome this limitation, several methods have been proposed. One such method is the CEP, which offers efficient and analytical updates.

In CEP, the approximate factor is assumed to be further factorized with respect to the variable groups $\{\boldsymbol{\theta}_1, \cdots, \boldsymbol{\theta}_M\}$, which can be expressed as

$$\tilde{f}_i(\boldsymbol{\theta}) = \prod_m \tilde{f}_{im}(\boldsymbol{\theta}_m), \tag{10}$$

where $\{\tilde{f}_{im}(\boldsymbol{\theta}_m)\}_{m=1}^M$ are constrained to be in the exponential family[3]. As a result, the approximate posterior $q(\boldsymbol{\theta}_m)$ and the calibrating distribution $q^{\backslash i}(\boldsymbol{\theta})$ are both factorized over the variable groups. It is worth noting that the factorized message factors are also widely used in EP algorithms for large-scale applications.

Given the factorized form in (10), the objective is to update each subfactor $\tilde{f}_{im}(\boldsymbol{\theta}_m)$. By utilizing the law of iterated expectations, the moment $\mathbb{E}_{\hat{p}_i(\boldsymbol{\theta})}[\boldsymbol{\phi}(\boldsymbol{\theta}_m)]$ required for updating $\tilde{f}_{im}(\boldsymbol{\theta}_m)$ can be expressed as (Wang & Zhe, 2020)

$$\mathbb{E}_{\hat{p}_i(\boldsymbol{\theta})}[\boldsymbol{\phi}(\boldsymbol{\theta}_m)] = \mathbb{E}_{\hat{p}_i(\boldsymbol{\theta}_{\backslash m})}\left[\mathbb{E}_{\hat{p}_i(\boldsymbol{\theta}_m|\boldsymbol{\theta}_{\backslash m})}[\boldsymbol{\phi}(\boldsymbol{\theta}_m)]\right], \tag{11}$$

where $\hat{p}_i(\boldsymbol{\theta}_m|\boldsymbol{\theta}_{\backslash m})$ is the conditional distribution and $\hat{p}_i(\boldsymbol{\theta}_{\backslash m})$ is the marginal distribution. The conditional moment $\mathbb{E}_{\hat{p}_i(\boldsymbol{\theta}_m|\boldsymbol{\theta}_{\backslash m})}[\boldsymbol{\phi}(\boldsymbol{\theta}_m)]$ often has an analytical form since the rest of the variables except $\boldsymbol{\theta}_m$ are fixed. More generally, the conditional moment can be computed using quadrature methods. (Wang & Zhe, 2020)

To compute the moment in (11), EP requires the computation of the expectation of the conditional moment with respect to the marginal posterior $\hat{p}_i(\boldsymbol{\theta}_{\backslash m})$. However, this computation is also intractable for models with complex likelihoods. To address this challenge, CEP assumes that $q(\boldsymbol{\theta}_{\backslash m})$ and $\hat{p}_i(\boldsymbol{\theta}_{\backslash m})$ are close in high-density regions as their moments are matched (Wang & Zhe, 2020). In the sequel, CEP employs $q(\boldsymbol{\theta}_{\backslash m})$ as a surrogate for $\hat{p}_i(\boldsymbol{\theta}_{\backslash m})$ in the respective computation. The goal now becomes to calculate the expectation $\mathbb{E}_{q(\boldsymbol{\theta}_{\backslash m})}\left[\mathbb{E}_{\hat{p}_i(\boldsymbol{\theta}_m|\boldsymbol{\theta}_{\backslash m})}[\boldsymbol{\phi}(\boldsymbol{\theta}_m)]\right]$.

Note that the conditional moment $\mathbb{E}_{\hat{p}_i(\boldsymbol{\theta}_m|\boldsymbol{\theta}_{\backslash m})}[\boldsymbol{\phi}(\boldsymbol{\theta}_m)]$ is a function of the sufficient statistics of $\boldsymbol{\theta}_{\backslash m}$, denoted as $\mathbb{E}_{\hat{p}_i(\boldsymbol{\theta}_m|\boldsymbol{\theta}_{\backslash m})}[\boldsymbol{\phi}(\boldsymbol{\theta}_m)] = h(\boldsymbol{\phi}(\boldsymbol{\theta}_{\backslash m}))$, where $\boldsymbol{\phi}(\boldsymbol{\theta}_{\backslash m}) = \{\boldsymbol{\phi}(\boldsymbol{\theta}_1), \cdots, \boldsymbol{\phi}(\boldsymbol{\theta}_{m-1}), \boldsymbol{\phi}(\boldsymbol{\theta}_{m+1}), \cdots, \boldsymbol{\phi}(\boldsymbol{\theta}_M)\}$ is the set of sufficient statistics. This holds whenever the conditional density $\hat{p}_i(\boldsymbol{\theta}_m|\boldsymbol{\theta}_{\backslash m})$ depends on $\boldsymbol{\theta}_{\backslash m}$ only through the sufficient statistics $\boldsymbol{\phi}(\boldsymbol{\theta}_{\backslash m})$. In conjugate-exponential models, which we assume in our main results (Section 3), this property is automatically satisfied, since the log-conditional $\ln p(\boldsymbol{\theta}_m|\boldsymbol{\theta}_{\backslash m})$ is a multi-linear function of the sufficient statistics of $\boldsymbol{\theta}_m$ and $\boldsymbol{\theta}_{\backslash m}$ (Winn et al., 2005). If the expectation $\mathbb{E}_{q(\boldsymbol{\theta}_{\backslash m})}[h(\boldsymbol{\phi}(\boldsymbol{\theta}_{\backslash m}))]$ remains intractable, it can be approximated using the *delta approximation method* as $h(\mathbb{E}_{q(\boldsymbol{\theta}_{\backslash m})}[\boldsymbol{\phi}(\boldsymbol{\theta}_{\backslash m})])$. The delta approximation method can be interpreted as a special case of the Laplace approximation or, equivalently, as a first-order Taylor expansion around the mean, as discussed in Appendix C. Similar to expectation propagation (EP), the messages can be computed in a distributed fashion. However,

---

[2]By local refinement, we mean that each factor is updated individually based on its local context in the model, without requiring access to the entire joint distribution.

[3]We use the subscript $im$ to distinguish the group-specific subfactor $\tilde{f}_{im}(\boldsymbol{\theta}_m)$ from the full factor $\tilde{f}_i(\boldsymbol{\theta})$.

the convergence of the overall procedure remains an open problem. The full algorithm for CEP is provided in Appendix A.

## 2.3 Assumed Density Filtering

ADF is an online Bayesian inference method that can be seen as a special case of EP. It provides an efficient approach for approximating posterior distributions in a sequential manner. ADF is obtained by initializing all the approximating factors, except the first one, to unity and then updating each factor once in a single pass. The ADF algorithm shares similarities with EP but it simplifies certain aspects of the process. Particularly, in ADF, the removal step, which involves creating a calibrating distribution by removing a factor from the approximation, is ignored. Instead, the calibrating distribution is replaced by the full approximation, given by (Li et al., 2015)

$$q^{\backslash i}(\boldsymbol{\theta}) = q(\boldsymbol{\theta}).$$

Consequently, the tilted distribution in ADF can be expressed as:

$$\hat{p}_i(\boldsymbol{\theta}) \propto f_i(\boldsymbol{\theta})q(\boldsymbol{\theta}).$$

The subsequent steps in ADF are the same as in EP.

## 2.4 Connections and Differences

As discussed above, ABI methods are commonly categorized into two groups: VI-based methods (e.g., VI, VMP) and EP-based methods (e.g., EP, ADF, CEP). Both aim to approximate the posterior by minimizing a divergence between the true and approximate distributions, but they differ in the choice of divergence and resulting algorithmic behavior.

VI-based methods minimize $\mathrm{KL}(q\|p)$, where $q$ is the variational approximation and $p$ is the true posterior. This leads to algorithms that monotonically improve the ELBO and converge to a local optimum. However, such minimization typically favors compact approximations, often capturing only one mode of a multimodal posterior (Bishop, 2006).

In contrast, EP-based methods minimize $\mathrm{KL}(p\|q)$ in a local, factor-wise manner. This tends to yield approximations that better reflect the full support of the posterior (Bishop, 2006). EP relies on local message passing over factor graphs and can be extended to distributed or online versions. However, its convergence is not theoretically guaranteed, and moment computations involved in updates may be intractable in practice. While existing works demonstrate that VI and EP can be viewed as minimizing different cases of a generalized divergence, they do not explicitly reveal how the specific update rules of distinct algorithms relate to one another at the algorithmic level.

Prior to this work, practical studies have primarily focused on connections within each family, such as between VI and VMP, or between EP and its variants like ADF and CEP. The algorithmic relationships across the two families, particularly between VMP and CEP, have remained largely unexplored. Fig. 1 provides an overview of the landscape: the top and bottom parts represent VI- and EP-based methods, respectively. Black arrows indicate previously known connections, while red arrows highlight the new algorithmic links identified in this work, as detailed in Section 1.

## 3 Main Results

The previous section reviewed several representative approximate Bayesian inference methods, emphasizing the roles of the mean-field assumption and moment matching. To support the theoretical development, Appendix B presents two key lemmas that characterize the behavior of KL divergence minimization under different model assumptions. These lemmas provide the foundation for the analysis in this section.

We now present the main theoretical results of the paper. The first subsection introduces our central theorem, which establishes a connection between VMP and CEP under mild conditions. The second subsection

extends this result and explores its implications for other inference algorithms. The third subsection offers a probabilistic interpretation of the theorem, providing complementary insights to aid understanding. The final subsection summarizes the results and discusses practical considerations for implementation.

### 3.1 Connection between VMP and CEP

To establish the connection between CEP and VMP, we initially present the following lemma.

**Lemma 3:** *Assume $p(\boldsymbol{\theta})$ is a fixed distribution and $q(\boldsymbol{\theta})$ factorizes with respect to variable groups, i.e.,*

$$q(\boldsymbol{\theta}) = \prod_m q(\boldsymbol{\theta}_m),$$

*where each factor $q(\boldsymbol{\theta}_m)$ belongs to the exponential family. Then minimizing the divergence $KL(p\|q)$ with respect to $q$ gives*

$$\mathbb{E}_{q(\boldsymbol{\theta})}[\phi(\boldsymbol{\theta}_m)] = \mathbb{E}_{p(\boldsymbol{\theta})}[\phi(\boldsymbol{\theta}_m)], \forall m, \tag{12}$$

*where $\phi(\boldsymbol{\theta}_m)$ is the sufficient statistics of $q(\boldsymbol{\theta}_m)$.*

*Proof:* See Appendix E.

Lemma 3 can be seen as a combination of Lemma 1 and Lemma 2, establishing a connection between the conditional moment matching and the minimization of KL divergence. In CEP, the optimal factor is given by

$$\tilde{f}_{im}(\boldsymbol{\theta}_m) \propto q^{\natural}(\boldsymbol{\theta}_m)/q^{\backslash i}(\boldsymbol{\theta}_m),$$

where the calibrating distribution $q^{\backslash i}(\boldsymbol{\theta}_m)$ is as defined in Section 2 and the variational distribution $q^{\natural}(\boldsymbol{\theta})$ is obtained through moment matching, satisfying

$$\mathbb{E}_{q^{\natural}(\boldsymbol{\theta})}[\phi(\boldsymbol{\theta}_m)] = \mathbb{E}_{\hat{p}_i(\boldsymbol{\theta})}[\phi(\boldsymbol{\theta}_m)]. \tag{13}$$

To establish the connection between CEP and VMP, it is necessary to derive an analytical form for the factor $\tilde{f}_{im}(\boldsymbol{\theta}_m)$. Generally, $\tilde{f}_{im}(\boldsymbol{\theta}_m)$ does not possess an analytical form due to the involvement of moment matching in the computation of $q^{\natural}(\boldsymbol{\theta})$. However, by comparing (12) and (13), we can show that for conjugate-exponential models under mild conditions, $\tilde{f}_{im}(\boldsymbol{\theta}_m)$ does indeed have an analytical form.

Before stating this result, we introduce the following notation. We write $\hat{p}_i(\boldsymbol{\theta}_m|\mathbb{E}_q[\phi(\boldsymbol{\theta}_{\backslash m})])$ to denote the conditional distribution $\hat{p}_i(\boldsymbol{\theta}_m|\boldsymbol{\theta}_{\backslash m})$ with the sufficient statistics $\phi(\boldsymbol{\theta}_{\backslash m})$ replaced by their expectations $\mathbb{E}_q[\phi(\boldsymbol{\theta}_{\backslash m})]$ under the current approximate posterior $q$. In the exponential family form, this corresponds to evaluating the natural parameters at $\boldsymbol{\eta}_m(\mathbb{E}_q[\phi(\boldsymbol{\theta}_{\backslash m})])$.

**Lemma 4:** *Consider a conjugate-exponential probabilistic model represented as a Bayesian network. If the expectations are calculated using the delta approximation method, the optimal factor in CEP is expressed as*

$$\tilde{f}_{im}(\boldsymbol{\theta}_m) \propto \frac{\hat{p}_i(\boldsymbol{\theta}_m|\mathbb{E}_q[\phi(\boldsymbol{\theta}_{\backslash m})])}{q^{\backslash i}(\boldsymbol{\theta}_m)}. \tag{14}$$

*Proof:* See Appendix E.

Based on the analytical form of $\tilde{f}_{im}(\boldsymbol{\theta}_m)$, we can show the connection between the CEP and VMP, and state the following theorem.

**Theorem 1:** *Consider a conjugate-exponential probabilistic model represented as a Bayesian network. Suppose the variational distribution follows the mean-field assumption and the observations are i.i.d. Then the CEP and VMP yield the same update equations under the following conditions:*

- *The update in CEP is performed on the variable groups.*

- *The expectations in CEP are calculated using the delta approximation method.*

*Proof:* To prove Theorem 1, we first give the following lemma.

**Lemma 5:** *Consider a conjugate-exponential probabilistic model represented as a Bayesian network. Suppose the variational distribution follows the mean-field assumption and the observations are i.i.d. If the update in CEP is performed on the variable groups, then a sufficient condition for the equivalence between the update equations of CEP and VMP is*

$$\ln \tilde{f}_{im}(\boldsymbol{\theta}_m) = \mathbb{E}_{q(\boldsymbol{\theta}_{\backslash m})}[\ln f_i(\boldsymbol{\theta})]. \tag{15}$$

*Proof:* To prove this lemma, we start by considering the logarithm of the optimal distribution in VMP, given by:

$$\begin{aligned}
\ln q^*(\boldsymbol{\theta}_m) &= \mathbb{E}_{q(\boldsymbol{\theta}_{\backslash m})}[\ln p(\boldsymbol{\theta}, \mathcal{D})] + \text{const4} \\
&= \mathbb{E}_{q(\boldsymbol{\theta}_{\backslash m})}\Big[\sum_i \ln f_i(\boldsymbol{\theta})\Big] + \text{const4} \\
&= \sum_i \mathbb{E}_{q(\boldsymbol{\theta}_{\backslash m})}[\ln f_i(\boldsymbol{\theta})] + \text{const4}.
\end{aligned} \tag{16}$$

where $p(\boldsymbol{\theta}, \mathcal{D}) = \prod_i f_i(\boldsymbol{\theta})$ follows from (7) under the i.i.d. assumption. Note that we also use $f_i(\boldsymbol{\theta})$ to represent the likelihood and prior, as in CEP. On the other hand, if the update in CEP is performed on the variable groups, then the optimal distribution for each variable group can be expressed as the product of the approximate factors:[4]

$$q^*(\boldsymbol{\theta}_m) \propto \prod_i \tilde{f}_{im}(\boldsymbol{\theta}_m).$$

Taking the logarithm of both sides gives:

$$\ln q^*(\boldsymbol{\theta}_m) = \sum_i \ln \tilde{f}_{im}(\boldsymbol{\theta}_m) + \text{constant5}. \tag{17}$$

By comparing (16) and (17), it can be seen that the updates of CEP and VMP are the same if (15) holds. ∎

Then we show that (15) in Lemma 5 holds if the expectations are approximated using the delta approximation method. From Lemma 4, the logarithm of message factor $\tilde{f}_{im}(\boldsymbol{\theta}_m)$ in CEP can be represented as

$$\ln \tilde{f}_{im}(\boldsymbol{\theta}_m) = \ln \hat{p}_i(\boldsymbol{\theta}_m | \mathbb{E}_q[\boldsymbol{\phi}(\boldsymbol{\theta}_{\backslash m})]) - \ln q^{\backslash i}(\boldsymbol{\theta}_m). \tag{18}$$

From (16), it can be seen that the optimal variational distribution in VMP consists of some independent terms, which can be expressed as

$$\begin{aligned}
\mathbb{E}_{q(\boldsymbol{\theta}_{\backslash m})}[\ln f_i(\boldsymbol{\theta})] &= \mathbb{E}_{q(\boldsymbol{\theta}_{\backslash m})}[\ln f_i(\boldsymbol{\theta})] + \ln q^{\backslash i}(\boldsymbol{\theta}_m) - \ln q^{\backslash i}(\boldsymbol{\theta}_m) \\
&= \mathbb{E}_{q(\boldsymbol{\theta}_{\backslash m})}[\ln f_i(\boldsymbol{\theta}) q^{\backslash i}(\boldsymbol{\theta}_m)] - \ln q^{\backslash i}(\boldsymbol{\theta}_m) \\
&= \mathbb{E}_{q(\boldsymbol{\theta}_{\backslash m})}[\ln \hat{p}_i(\boldsymbol{\theta}_m | \boldsymbol{\theta}_{\backslash m})] - \ln q^{\backslash i}(\boldsymbol{\theta}_m).
\end{aligned} \tag{19}$$

Comparing (18) and (19), we see that equation (15) holds if and only if

$$\mathbb{E}_{q(\boldsymbol{\theta}_{\backslash m})}[\ln \hat{p}_i(\boldsymbol{\theta}_m | \boldsymbol{\theta}_{\backslash m})] = \ln \hat{p}_i(\boldsymbol{\theta}_m | \mathbb{E}_q[\boldsymbol{\phi}(\boldsymbol{\theta}_{\backslash m})]).$$

---

[4]Here, we also use the notation $q^*(\boldsymbol{\theta}_m)$ to denote the optimal variational distribution in CEP.

Now, we show that this equality holds under the conditions in Theorem 1. Since the model is conjugate-exponential, the conditional distribution $\hat{p}_i(\boldsymbol{\theta}_m|\boldsymbol{\theta}_{\backslash m})$ is in the exponential family and can be expressed as:

$$\hat{p}_i(\boldsymbol{\theta}_m|\boldsymbol{\theta}_{\backslash m}) = h(\boldsymbol{\theta}_m) \exp\left\{\boldsymbol{\eta}_m(\boldsymbol{\theta}_{\backslash m})^T \boldsymbol{\phi}(\boldsymbol{\theta}_m) - Z_m(\boldsymbol{\eta}_m(\boldsymbol{\theta}_{\backslash m}))\right\}.$$

where $\boldsymbol{\phi}(\boldsymbol{\theta}_m)$ is the vector of sufficient statistics; $\boldsymbol{\eta}_m$ are the natural parameters; and $Z_m(\cdot)$ is the log partition function. Its logarithm can be expressed by

$$\ln \hat{p}_i(\boldsymbol{\theta}_m|\boldsymbol{\theta}_{\backslash m}) = \ln h(\boldsymbol{\theta}_m) + \boldsymbol{\eta}_m(\boldsymbol{\theta}_{\backslash m})^T \boldsymbol{\phi}(\boldsymbol{\theta}_m) - Z_m(\boldsymbol{\eta}_m(\boldsymbol{\theta}_{\backslash m})).$$

Taking expectation with respect to $q(\boldsymbol{\theta}_{\backslash m})$ yields

$$\mathbb{E}_{q(\boldsymbol{\theta}_{\backslash m})}[\ln \hat{p}_i(\boldsymbol{\theta}_m|\boldsymbol{\theta}_{\backslash m})] = \ln h(\boldsymbol{\theta}_m) + \mathbb{E}_{q(\boldsymbol{\theta}_{\backslash m})}[\boldsymbol{\eta}_m(\boldsymbol{\theta}_{\backslash m})]^T \boldsymbol{\phi}(\boldsymbol{\theta}_m) + \text{const6}.$$

For a conjugate-exponential model, $\ln \hat{p}_i(\boldsymbol{\theta}_m|\boldsymbol{\theta}_{\backslash m})$ is a *multi-linear* function of the sufficient statistics of $\boldsymbol{\theta}_m$ and the variables in $\boldsymbol{\theta}_{\backslash m}$ (i.e., $\boldsymbol{\phi}(\boldsymbol{\theta}_j), \forall j = 1, \cdots, M$). As a result, $\boldsymbol{\eta}_m(\boldsymbol{\theta}_{\backslash m})$ is a multi-linear function of $\boldsymbol{\phi}(\boldsymbol{\theta}_j), \forall j = 1, \cdots, M$, and we have

$$\begin{aligned}
\mathbb{E}_{q(\boldsymbol{\theta}_{\backslash m})}[\ln \hat{p}_i(\boldsymbol{\theta}_m|\boldsymbol{\theta}_{\backslash m})] &= \ln h(\boldsymbol{\theta}_m) + \mathbb{E}_{q(\boldsymbol{\theta}_{\backslash m})}[\boldsymbol{\eta}_m(\boldsymbol{\phi}(\boldsymbol{\theta}_{\backslash m}))]^T \boldsymbol{\phi}(\boldsymbol{\theta}_m) + \text{const6} \\
&= \ln h(\boldsymbol{\theta}_m) + \boldsymbol{\eta}_m(\mathbb{E}_q[\boldsymbol{\phi}(\boldsymbol{\theta}_{\backslash m})])^T \boldsymbol{\phi}(\boldsymbol{\theta}_m) + \text{const6} \\
&= \ln \hat{p}_i(\boldsymbol{\theta}_m|\mathbb{E}_q[\boldsymbol{\phi}(\boldsymbol{\theta}_{\backslash m})]).
\end{aligned} \tag{20}$$

Thus, equation (15) holds, and by Lemma 5, we conclude that the updates of CEP and VMP are equivalent under the specified conditions. ∎

In practical applications, these preconditions and conditions are often satisfied, as demonstrated in the example in Section 4. Below, we discuss their implications.

The i.i.d. assumption enables the VMP update to be expressed as a summation of $N$ terms, each corresponding to a factor in the CEP framework. This means the VMP update can be interpreted as merging messages from data nodes, which will be discussed further in the next subsection. The conjugate-exponential condition ensures that the factor updates in CEP have analytical forms. When this assumption does not hold, the direct connection between VMP and CEP may break down.

Regarding the specific conditions, the first ensures that $q(\boldsymbol{\theta}_m)$ in CEP is expressed as a product of factors, enabling parallel computation. The second simplifies the expectation calculations in CEP. For models of practical interest, the outer expectation in CEP is typically intractable, making this approximation both necessary and effective (Pan et al., 2020; Fang et al., 2021a).

Note that equation (20) can be seen as a reparameterization process, where the statistic parameters of $\boldsymbol{\theta}_{\backslash m}$ are replaced with their expectations regarding the corresponding distribution. This procedure is thoroughly discussed in the original VMP paper, along with a simple illustrative example (Winn et al., 2005). In particular, this reparameterization is exact (i.e., no approximation error is introduced) precisely because the conjugate-exponential structure guarantees that $\ln p(\boldsymbol{\theta}_m|\boldsymbol{\theta}_{\backslash m})$ is a multi-linear function of the sufficient statistics $\boldsymbol{\phi}(\boldsymbol{\theta}_m)$ and $\boldsymbol{\phi}(\boldsymbol{\theta}_{\backslash m})$ (Winn et al., 2005), so that $\mathbb{E}_{q(\boldsymbol{\theta}_{\backslash m})}[\boldsymbol{\eta}_m(\boldsymbol{\phi}(\boldsymbol{\theta}_{\backslash m}))] = \boldsymbol{\eta}_m(\mathbb{E}_q[\boldsymbol{\phi}(\boldsymbol{\theta}_{\backslash m})])$ holds without invoking the delta approximation.

It is also worth noting that the connection between CEP and VMP can be viewed from a more general perspective. To see this, note that a fundamental assumption in CEP is that the message factor $\tilde{f}_i$ factorizes with respect to variable groups, allowing the approximate posterior to be expressed as

$$q(\boldsymbol{\theta}) \propto \prod_i \tilde{f}_i(\boldsymbol{\theta}) = \prod_i \prod_m \tilde{f}_{im}(\boldsymbol{\theta}_m) = \prod_m q(\boldsymbol{\theta}_m),$$

which in fact corresponds to the mean-field assumption. From Lemma 2, we know that the optimal solution of $\min_{q(\boldsymbol{\theta}_m)} \mathrm{KL}(p(\boldsymbol{\theta}|\mathcal{D}) \| q(\boldsymbol{\theta}))$ is given by $q^*(\boldsymbol{\theta}_m) = p(\boldsymbol{\theta}_m|\mathcal{D})$, which can be further written as

$$
\begin{aligned}
q^*(\boldsymbol{\theta}_m) &= p(\boldsymbol{\theta}_m|\mathcal{D}) \\
&= \int p(\boldsymbol{\theta}_m, \boldsymbol{\theta}_{\backslash m}|\mathcal{D}) d\boldsymbol{\theta}_{\backslash m} \\
&= \int p(\boldsymbol{\theta}_m|\boldsymbol{\theta}_{\backslash m}, \mathcal{D}) p(\boldsymbol{\theta}_{\backslash m}|\mathcal{D}) d\boldsymbol{\theta}_{\backslash m} \\
&= \mathbb{E}_p[p(\boldsymbol{\theta}_m|\boldsymbol{\theta}_{\backslash m}, \mathcal{D})].
\end{aligned}
$$

By applying the approximations in CEP, i.e., using $q(\boldsymbol{\theta}_{\backslash m})$ as a surrogate of $p(\boldsymbol{\theta}_{\backslash m}|\mathcal{D})$ in moment computation and the delta approximation, the optimal variational distribution can be approximated as

$$
q^*(\boldsymbol{\theta}_m) = \mathbb{E}_p[p(\boldsymbol{\theta}_m|\boldsymbol{\theta}_{\backslash m}, \mathcal{D})] \approx p(\boldsymbol{\theta}_m|\mathbb{E}_q[\boldsymbol{\phi}(\boldsymbol{\theta}_{\backslash m})], \mathcal{D}). \tag{21}
$$

In VMP, each optimal variational distribution becomes

$$
\begin{aligned}
\ln q^*(\boldsymbol{\theta}_m) &= \mathbb{E}_{q(\boldsymbol{\theta}_{\backslash m})}[\ln p(\boldsymbol{\theta}, \mathcal{D})] + \mathrm{constant2} \\
&= \mathbb{E}_{q(\boldsymbol{\theta}_{\backslash m})}[\ln p(\boldsymbol{\theta}_m|\boldsymbol{\theta}_{\backslash m}, \mathcal{D})] + \mathrm{constant3} \\
&= \ln p(\boldsymbol{\theta}_m|\mathbb{E}_q[\boldsymbol{\phi}(\boldsymbol{\theta}_{\backslash m})], \mathcal{D}),
\end{aligned}
$$

which leads to

$$
q^*(\boldsymbol{\theta}_m) = p(\boldsymbol{\theta}_m|\mathbb{E}_q[\boldsymbol{\phi}(\boldsymbol{\theta}_{\backslash m})], \mathcal{D}). \tag{22}
$$

By comparing (21) and (22), it can be seen that the inherent objective of both VMP and CEP is to approximate the conditional marginal distribution $p(\boldsymbol{\theta}_m|\mathbb{E}_q[\boldsymbol{\phi}(\boldsymbol{\theta}_{\backslash m})], \mathcal{D})$. VMP and CEP start from different KL formulations. However, in both cases, the theoretical optimal is the same due to the properties of the KL. This justifies the derived connection.

## 3.2 Extensions and Implications

The previous section established a strong connection between CEP and VMP. Expanding on this connection, we present new theoretical results regarding the convergence and scalability of several ABI methods.

### 3.2.1 Convergence of CEP

As previously mentioned, the convergence of EP is not generally guaranteed. To address this issue, some approaches apply energy optimization techniques directly to the associated objective function rather than relying on local updates. For instance, they implement EP based on the convergent double-loop optimization algorithm (Opper et al., 2005; Hasenclever et al., 2017). However, these approaches require additional designs and exhibit increased computational complexities.

Since CEP is developed from EP, its convergence properties also remain an open question. Nevertheless, by leveraging the established connection with VMP, we can demonstrate that CEP is guaranteed to converge under certain mild conditions. Specifically, we present the following corollary.

**Corollary 1:** *Consider a conjugate-exponential probabilistic model represented as a Bayesian network. Suppose the variational distribution follows the mean-field assumption and the observations are i.i.d. If the conditions in Theorem 1 hold, then CEP updates are guaranteed to converge to a local minimum of the KL divergence.*

*Proof:* See Appendix E.

From (16), the optimal variational distribution $q^*(\boldsymbol{\theta}_m)$ in VMP is

$$
\ln q^*(\boldsymbol{\theta}_m) = \sum_i \mathbb{E}_{q(\boldsymbol{\theta}_{\backslash m})}[\ln f_i(\boldsymbol{\theta})] + \mathrm{const4}.
$$

As shown in Section 2, this update increases the ELBO at each iteration, ensuring the convergence property of VMP. According to Lemma 5, we have $\mathbb{E}_{q(\boldsymbol{\theta}_{\backslash m})}[\ln f_i(\boldsymbol{\theta})] = \ln \tilde{f}_{im}(\boldsymbol{\theta}_m)$. Since $\tilde{f}_{im}(\boldsymbol{\theta}_m)$ is the message factor, the term $\mathbb{E}_{q(\boldsymbol{\theta}_{\backslash m})}[\ln f_i(\boldsymbol{\theta})]$ can be interpreted as the message sent from the $i$th data node. Thus, the update in VMP can be viewed as merging all the messages sent by data nodes. In other words, in CEP, merging the message factors $\tilde{f}_{im}(\boldsymbol{\theta}_m)$ sent by data nodes increases the ELBO, thereby ensuring convergence.

It is important to note that in the standard implementations of CEP, updates are performed on the factors instead of the variable groups. In other words, $\tilde{f}_i$ is updated sequentially in each iteration. This factor-based update mechanism allows for a more fine-grained local optimization, which might be the reason for its superior performance in various tasks. However, this type of local optimization does not guarantee convergence in general. If the updates in CEP are performed on the factors rather than on the variable groups, the convergence guarantee is lost.

To see this, note that if the updates in CEP are performed on the factors, the increase of ELBO is not guaranteed. Specifically, the ELBO can be expressed as

$$\mathcal{L} = \int \prod_m q(\boldsymbol{\theta}_m) \left\{ \ln p(\boldsymbol{\theta}, \mathcal{D}) - \sum_m \ln q(\boldsymbol{\theta}_m) \right\} d\boldsymbol{\theta}$$
$$= \int q(\boldsymbol{\theta}_m) \mathbb{E}_{q(\boldsymbol{\theta}_{\backslash m})}[\ln p(\boldsymbol{\theta}, \mathcal{D})] d\boldsymbol{\theta}_m - \int q(\boldsymbol{\theta}_m) \ln q(\boldsymbol{\theta}_m) d\boldsymbol{\theta}_m$$
$$= \int \prod_i \tilde{f}_{im}(\boldsymbol{\theta}_m) \sum_i \mathbb{E}_{q(\boldsymbol{\theta}_{\backslash m})}[f_i(\boldsymbol{\theta})] d\boldsymbol{\theta}_m + \int \prod_i \tilde{f}_{im}(\boldsymbol{\theta}_m) \sum_i \ln \tilde{f}_{im}(\boldsymbol{\theta}_m) d\boldsymbol{\theta}_m.$$

The optimal factor in CEP can be written as

$$\ln \tilde{f}_{im}(\boldsymbol{\theta}_m) = \ln \hat{p}_i(\boldsymbol{\theta}_m | \mathbb{E}_q[\boldsymbol{\phi}(\boldsymbol{\theta}_{\backslash m})]) - \ln q^{\backslash i}(\boldsymbol{\theta}_m)$$
$$= \ln q^{\backslash i}(\boldsymbol{\theta}_m) f_i(\boldsymbol{\theta}_m | \mathbb{E}_q[\boldsymbol{\phi}(\boldsymbol{\theta}_{\backslash m})]) - \ln q^{\backslash i}(\boldsymbol{\theta}_m)$$
$$= \ln f_i(\boldsymbol{\theta}_m | \mathbb{E}_q[\boldsymbol{\phi}(\boldsymbol{\theta}_{\backslash m})]),$$

which leads to $\tilde{f}_{im}(\boldsymbol{\theta}_m) = f_i(\boldsymbol{\theta}_m | \mathbb{E}_q[\boldsymbol{\phi}(\boldsymbol{\theta}_{\backslash m})])$. Due to the multiplication and integration involved in ELBO, optimizing $\mathcal{L}$ with respect to $\tilde{f}_{im}(\boldsymbol{\theta}_m)$ does not yield the same results as in CEP. Therefore, each update does not necessarily increase the ELBO, and CEP may not converge in this scenario. Similarly, this local optimization of message factors is also the reason why standard EP may not converge.

### 3.2.2 Connections to Streaming Bayes

The concept of EP is developed from ADF, an online Bayesian algorithm designed for streaming data. As CEP is a variant of EP, it can also be adapted into a streaming version, which we refer to as *streaming CEP*. Streaming CEP is obtained by applying the surrogate distribution and delta approximations of CEP to the ADF framework. Due to the established equivalence between CEP and VMP, a streaming version of VMP can be derived analogously. The resulting method shares a close connection with streaming variational Bayes (Broderick et al., 2013), although it is developed from a distinct perspective and offers different interpretations.

In Section II, we observe that ADF differs from EP in the factor removing step. In ADF, the removing step is ignored, and the calibrating distribution is replaced by the full approximation obtained from the previous iteration. The updated approximating posterior is computed by directly multiplying the previous approximation with the newly updated message factor associated with the added data.

Mathematically, assuming the current approximation is denoted as $q(\boldsymbol{\theta})$, the new posterior in ADF is

$$q^*(\boldsymbol{\theta}) = \min_{\hat{q}(\boldsymbol{\theta})} \mathrm{KL}(\hat{p}_i(\boldsymbol{\theta}) \| \hat{q}(\boldsymbol{\theta})),$$

where $\hat{p}_i(\boldsymbol{\theta}) \propto f_i(\boldsymbol{\theta}) q(\boldsymbol{\theta})$. The resulting $q^*(\boldsymbol{\theta})$ is then used as the current approximation in the next iteration. In a conjugate-exponential model with mutually independent variable groups, the update of the posterior

for each variable has a closed-form solution, given by

$$q^*(\boldsymbol{\theta}_m) = \hat{p}_i(\boldsymbol{\theta}_m) = \mathbb{E}_{\hat{p}_i(\boldsymbol{\theta}_{\backslash m})}[\hat{p}_i(\boldsymbol{\theta}_m | \boldsymbol{\theta}_{\backslash m})].$$

Upon the arrival of new data, we can optimize each variable group and multiply their distributions together to obtain the new approximation $q^*(\boldsymbol{\theta})$.

As mentioned in the previous subsection, the variable update in VMP merges all the messages sent from the other nodes simultaneously. If the data arrives in a streaming manner, we can sequentially merge the messages to update the variables. Building upon this insight, we can easily modify the VMP to a streaming version. Specifically, when a new sample $\mathbf{x}_i$ arrives, the updated estimate of the posterior in VMP is

$$
\begin{aligned}
\ln q^*(\boldsymbol{\theta}_m) &= \mathbb{E}_{q(\boldsymbol{\theta}_{\backslash m})}[\ln p(\boldsymbol{\theta}, \mathcal{D})] + \text{const2} \qquad (23) \\
&= \mathbb{E}_{q(\boldsymbol{\theta}_{\backslash m})}[\ln f_i(\boldsymbol{\theta}) q(\boldsymbol{\theta})] + \text{const3} \\
&= \mathbb{E}_{q(\boldsymbol{\theta}_{\backslash m})}[\ln \hat{p}_i(\boldsymbol{\theta})] \\
&= \mathbb{E}_{q(\boldsymbol{\theta}_{\backslash m})}[\ln \hat{p}_i(\boldsymbol{\theta}_m | \boldsymbol{\theta}_{\backslash m})] \\
&= \ln \hat{p}_i(\boldsymbol{\theta}_m | \mathbb{E}_q[\boldsymbol{\phi}(\boldsymbol{\theta}_{\backslash m})])
\end{aligned}
$$

Here, the joint distribution can be expressed as $p(\boldsymbol{\theta}, \mathcal{D}) = \hat{p}_i(\boldsymbol{\theta}) \propto f_i(\boldsymbol{\theta}) q(\boldsymbol{\theta})$. Similar to ADF, we can optimize each variable group through (23) and then multiply the respective distributions together to obtain the new approximate estimate.

It is worth noting that the algorithm can be easily extended to scenarios where data arrive in a batch version. Additionally, standard VI with i.i.d. observations can also be easily modified to a streaming version through this framework. Moreover, it can be seen that the primary difference between streaming VMP and ADF lies in the computation of expectations: ADF computes expectations with respect to the marginal tilted distribution $\hat{p}_i(\boldsymbol{\theta}_{\backslash m})$, whereas streaming VMP uses the current approximation $q(\boldsymbol{\theta}_{\backslash m})$ and applies the delta approximation. When these two approximation steps are also applied to ADF, the resulting procedure coincides with streaming CEP. Based on this observation and the connection between VMP and CEP, we present the following corollary.

**Corollary 2:** *Consider a conjugate-exponential probabilistic model represented as a Bayesian network. Suppose the variational distribution follows the mean-field assumption and the observations are i.i.d. Then, streaming VMP and streaming CEP yield the same update equations, where streaming CEP is obtained from ADF by:*

- *using the current approximation $q(\boldsymbol{\theta}_{\backslash m})$ as a surrogate of $\hat{p}_i(\boldsymbol{\theta}_{\backslash m})$ in the computation of the expectation;*

- *computing the expectations using the delta approximation method.*

*Proof:* See Appendix E.
Since VMP is a special case of VI, it follows that streaming VI also yields the same update equations as streaming CEP under these conditions, provided that the underlying probabilistic model is a conjugate-exponential model.

Note that the streaming version of VMP or CEP performs a one-pass update, discarding the data once they are updated, which significantly reduces the storage requirements. Additionally, the variable update in VMP can be implemented in a distributed manner since the messages can also be calculated in parallel. The resulting algorithm is similar to the distributed VMP (Masegosa et al., 2016).

### 3.3 Interpretation via Graphical Models

Since both VMP and CEP are closely related to graphical models, we can gain further insights into their connection from a graphical model perspective. Specifically, we assume that the model takes the form of a Bayesian network, where the joint distribution factorizes as $p(\mathbf{V}) = \prod_i p(\mathbf{v}_i | \text{pa}_i)$, with $\mathbf{V} = \{\boldsymbol{\theta}, \mathcal{D}\}$ containing

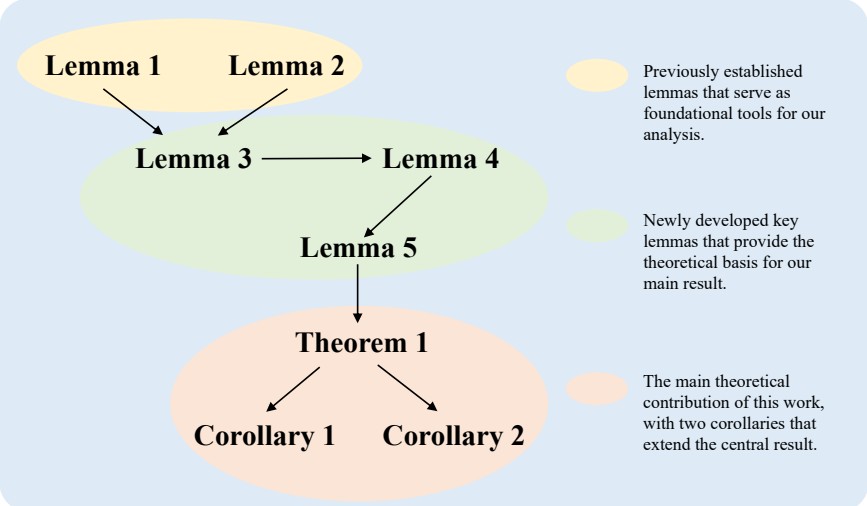

Figure 3: A summary of the theoretical results and their connections in this study.

all visible and hidden variables, $\mathrm{pa}_i$ denoting the parents of node $i$, and $\mathbf{v}_i$ the variable associated with node $i$ (see Fig. 2(a)). We adopt a similar notation to that used in the original VMP paper[5].

As shown in Appendix D, by exploiting the conjugate-exponential structure of the model, the natural parameter of the optimal variational distribution in VMP can be expressed as

$$\tilde{\boldsymbol{\eta}}_j^* = \tilde{\boldsymbol{\eta}}_j(\{\langle \boldsymbol{\phi}_s \rangle\}_{s \in \mathrm{pa}_j}) + \sum_{\mathbf{x}_k \in D} \tilde{\boldsymbol{\eta}}_{kj}(\mathbf{x}_k, \{\langle \boldsymbol{\phi}_t \rangle\}_{t \neq j}), \tag{24}$$

where $\langle \cdot \rangle$ denotes the expectation under $q$, $\mathrm{pa}_j$ is the parent set of node $j$, and $\tilde{\boldsymbol{\eta}}_j$ and $\tilde{\boldsymbol{\eta}}_{kj}$ are the reparameterized natural parameter functions evaluated at the expected sufficient statistics. The first term aggregates messages from the parent nodes, and the second term merges messages from all data nodes.

In CEP, using the factorized message structure and Lemma 5, the approximate factor from the $k$th data node can be expressed as (see Appendix D for the full derivation)

$$\ln \tilde{f}_k(\boldsymbol{\theta}_j) = \tilde{\boldsymbol{\eta}}_{kj}(\mathbf{x}_k, \{\langle \boldsymbol{\phi}_t \rangle\}_{t \neq j})^T \boldsymbol{\phi}_j(\boldsymbol{\theta}_j) + \lambda(\mathbf{x}_k, \{\langle \boldsymbol{\phi}_t \rangle\}_{t \neq j}), \tag{25}$$

where $\boldsymbol{\phi}_j(\boldsymbol{\theta}_j)$ is the sufficient statistics of node $j$ and $\lambda$ contains terms irrelevant to $\boldsymbol{\phi}_j(\boldsymbol{\theta}_j)$. Combining this with the prior term $\ln p(\boldsymbol{\theta}_j | \mathrm{pa}_j)$, the resulting distribution shares the same natural parameters as in (24), confirming the equivalence from the graphical model perspective.

Generally, the factors $\{\tilde{f}_k\}$ can be interpreted as the messages sent from the data nodes, after replacing the messages from the co-parent nodes with their expected sufficient statistics. Additionally, for a fully factorized model, the standard EP reduces to loopy belief propagation. More discussions concerning the performance and convergence of LBP can be found in Frey & MacKay (1997); Li et al. (2019); Du et al. (2018a).

### 3.4 Summary and Practical Suggestions

Fig. 3 summarizes the logical connections among the theoretical results developed in this work, from the foundational lemmas (Lemmas 1–3) through the closed-form message factor (Lemma 4) and the sufficient condition for equivalence (Lemma 5), to the central result (Theorem 1) and its two corollaries on convergence (Corollary 1) and streaming equivalence (Corollary 2). Below, we provide practical recommendations based on these results.

Based on the theoretical results developed in this work, we offer practical recommendations for selecting and applying different ABI methods:

---

[5]A more detailed discussion about the basics of graphical models can be found in Bishop (2006).

- *Convergent CEP*: As established in **Corollary 1**, the CEP algorithm is guaranteed to converge under the conditions specified in **Theorem 1**. This provides a clear guideline for safely applying CEP in practice without concern for convergence issues, which have traditionally been one of the main limitations of EP-based methods.

- *Streaming or parallel VMP*: The update steps of VMP can be interpreted as aggregating messages from other variable groups. This observation, supported by our theoretical analysis, shows that both algorithms are naturally compatible with streaming or parallel settings when the conditions in **Theorem 1** are met. In particular, (23) provides a closed-form formulation for constructing streaming variants of VMP by leveraging their structural similarities with CEP and ADF.

- *Streaming ABI from scratch*: When designing a streaming ABI algorithm from the ground up, a natural first step is to assess whether the model satisfies the conjugate-exponential assumption with independent variable partitions. If this condition holds, streaming VMP offers efficient closed-form updates. Otherwise, one may consider using moment matching to project the posterior distribution into a tractable exponential family form. Our results, particularly **Corollary 2**, provide a theoretical connection between streaming CEP and streaming VMP, offering a unified view that can guide algorithmic choices based on model structure and computational constraints.

While these suggestions represent direct applications of our theoretical findings, the broader insights revealed in this work open up new possibilities for designing more flexible and efficient Bayesian inference algorithms. We believe that exploring these directions, especially in the context of large-scale or streaming data, is a promising avenue for future research.

## 4 Example

In this section, we demonstrate the strong connections between the updates of VMP and CEP in the context of a Bayesian tensor decomposition model. Our emphasis is on the canonical polyadic decomposition (CPD), which is an essential technique in machine learning and has been used in various real-world applications. Our choice of Bayesian CPD as the illustrative example is motivated by its prominent role in the original CEP paper (Wang & Zhe, 2020), where the inference algorithm is extensively discussed. We start by introducing a probabilistic model for the CPD approach. To this end, we apply both VMP and CEP to infer the associated posterior distribution. We then extend the resulting algorithm to a streaming setting, enabling it to process sequentially arriving data. Finally, we evaluate the proposed method on an image completion task, demonstrating the equivalence of the two approaches and their effectiveness in practical scenarios. For notational convenience, throughout this section we use $\langle \cdot \rangle$ as shorthand for the expectation $\mathbb{E}_q[\cdot]$ under the current approximate posterior.

### 4.1 Probabilistic modeling

We denote a $K$-mode tensor by $\mathcal{X} \in \mathbb{R}^{d_1 \times \cdots \times d_K}$, where $d_k$ is the dimension of the $k$-th mode. The entry value at location $\mathbf{i} = (i_1, \cdots, i_K)$ is denoted as $x_\mathbf{i}$. To perform tensor decomposition, we introduce an $R$-dimensional embedding vector $\mathbf{u}_j^k$ to represent each object in mode $k$. Then, a $d_k \times R$ matrix can be constructed by stacking all the embedding vectors in mode $k$, i.e., $\mathbf{U}^k = [\mathbf{u}_1^k, \cdots, \mathbf{u}_{d_k}^k]^T$. Tensor decomposition aims to find the embedding matrices of all modes $\mathcal{U} = \{\mathbf{U}^1, \cdots, \mathbf{U}^K\}$ from the observed entries.

Mathematically, the CPD of a given tensor $\mathcal{X}$ is written as

$$\mathcal{X} = [\![\mathbf{U}^1, \cdots, \mathbf{U}^K]\!],$$

where $[\![\cdot]\!]$ is the Kruskal operator. A graphical illustration of a three-dimensional CPD ($K = 3$) is shown in Fig. 4. For each entry $x_\mathbf{i}$, we have

$$x_\mathbf{i} = \sum_{r=1}^{R} \prod_{k=1}^{K} u_{i_k, r}^k = \mathbf{1}^T(\mathbf{u}_{i_1}^1 \circ \cdots \circ \mathbf{u}_{i_K}^K),$$

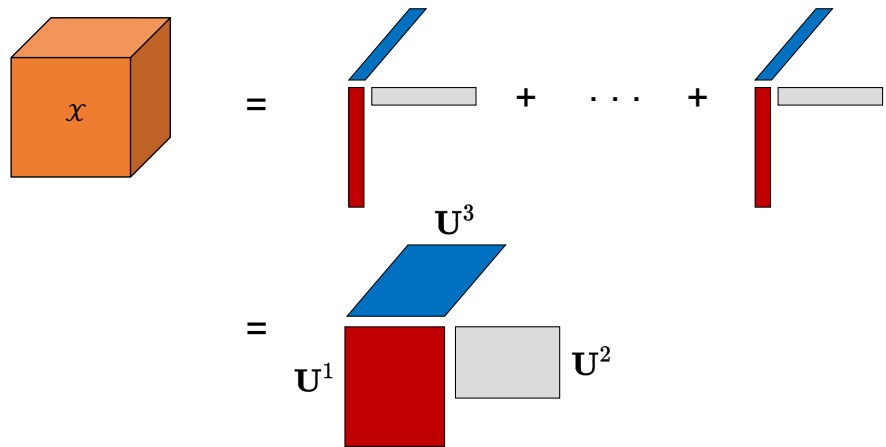

Figure 4: A graphical illustration of the three-dimensional CPD.

where $\circ$ is the Hadamard product.

Consider a $K$-mode tensor $\mathcal{Y}$ with $N$ observed entries denoted as $\{y_{\mathbf{i}}\}_{\mathbf{i}\in\mathcal{S}}$. Here, $\mathcal{S}$ represents the index set, and its cardinality is $|\mathcal{S}| = N$. We assume that the observations are contaminated with i.i.d. Gaussian noise. Then the likelihood can be expressed as

$$p(y_{\mathbf{i}}|\mathcal{U}, \tau) = \mathcal{N}(y_{\mathbf{i}}|\mathbf{1}^T(\mathbf{u}_{i_1}^1 \circ \cdots \circ \mathbf{u}_{i_K}^K), \tau^{-1}),$$

where $\tau$ is the noise precision. We further assign a conjugate Gamma prior over $\tau$, given by

$$p(\tau|a_0, b_0) = \mathrm{Gam}(\tau|a_0, b_0).$$

For each embedding vector $\mathbf{u}_s^k$, we assign a Gaussian prior with mean $\boldsymbol{\beta}_s^k$ and covariance $v\mathbf{I}$, given by

$$p(\mathcal{U}) = \prod_{k=1}^{K}\prod_{s=1}^{d_k}\mathcal{N}(\mathbf{u}_s^k|\boldsymbol{\beta}_s^k, v\mathbf{I}),$$

where $\{\boldsymbol{\beta}_s^k\}$ and $v$ are pre-defined hyperparameters.

Consequently, the joint probability distribution is

$$p(\{y_{\mathbf{i}}\}_{\mathbf{i}\in\mathcal{S}}, \mathcal{U}, \tau) = \mathrm{Gam}(\tau|a_0, b_0)\prod_{k=1}^{K}\prod_{s=1}^{d_k}\mathcal{N}(\mathbf{u}_s^k|\boldsymbol{\beta}_s^k, v\mathbf{I}) \tag{26}$$
$$\cdot \prod_{\mathbf{i}\in\mathcal{S}}\mathcal{N}(y_{\mathbf{i}}|\mathbf{1}^T(\mathbf{u}_{i_1}^1 \circ \cdots \circ \mathbf{u}_{i_K}^K), \tau^{-1}).$$

Note that the prior and likelihood are conjugate and belong to the exponential family, thus the probabilistic model is a conjugate-exponential model. Additionally, the observations are assumed to be i.i.d., therefore satisfying the conditions in Theorem 1.

### 4.2 VMP

In VMP, the variables are assumed to be mutually independent, allowing us to factorize the variational distribution as

$$q(\mathcal{U}, \tau) = q(\tau)\prod_{k=1}^{K}\prod_{s=1}^{d_k}q(\mathbf{u}_s^k).$$

Since the probabilistic model is conjugate-exponential, the variational distribution for each variable is identical to its prior distribution. Consequently, the variational distribution is parameterized by

$$q(\mathcal{U}, \tau) = \mathrm{Gam}(\tau|a, b) \prod_{k=1}^{K} \prod_{s=1}^{d_k} \mathcal{N}(\mathbf{u}_s^k | \boldsymbol{\mu}_s^k, \boldsymbol{\Sigma}_s^k).$$

Due to the conjugacy property, we can derive closed-form updates for each variable. Here we present the key steps and leave the detailed derivation in Appendix F. Specifically, the optimal variational distribution of $\mathbf{u}_s^k$ is given by

$$q^*(\mathbf{u}_s^k) = \mathcal{N}(\mathbf{u}_s^k | \boldsymbol{\mu}_s^{k*}, \boldsymbol{\Sigma}_s^{k*}),$$

with the mean $\boldsymbol{\mu}_s^{k*}$ and covariance $\boldsymbol{\Sigma}_s^{k*}$ given by

$$\boldsymbol{\mu}_s^{k*} = \boldsymbol{\Sigma}_s^{k*} \left( \langle \tau \rangle \sum_{\mathbf{i} \in \mathcal{S}, i_k = s} y_{\mathbf{i}} \langle \mathbf{z}_{\mathbf{i}}^{\backslash k} \rangle + v \boldsymbol{\beta}_s^k \right), \tag{27}$$

$$\boldsymbol{\Sigma}_s^{k*} = \left( \langle \tau \rangle \sum_{\mathbf{i} \in \mathcal{S}, i_k = s} \langle \mathbf{z}_{\mathbf{i}}^{\backslash k} \mathbf{z}_{\mathbf{i}}^{\backslash k T} \rangle + v \mathbf{I} \right)^{-1},$$

where

$$\mathbf{z}_{\mathbf{i}}^{\backslash k} = \mathbf{u}_{i_1}^1 \circ \cdots \circ \mathbf{u}_{i_{k-1}}^{k-1} \circ \mathbf{u}_{i_{k+1}}^{k+1} \circ \cdots \circ \mathbf{u}_{i_K}^K.$$

The optimal variational distribution of noise precision $\tau$ is given by

$$q^*(\tau) = \mathrm{Gam}(\tau|a^*, b^*),$$

with $a^*$ and $b^*$ computed as follows

$$a^* = a_0 + \frac{N}{2},$$

$$b^* = b_0 + \frac{1}{2} \sum_{\mathbf{i} \in \mathcal{S}} [y_{\mathbf{i}}^2 - 2 y_{\mathbf{i}} \langle \mathbf{1}^T \mathbf{z}_{\mathbf{i}} \rangle + \langle (\mathbf{1}^T \mathbf{z}_{\mathbf{i}})^2 \rangle],$$

where $\mathbf{z}_{\mathbf{i}} = \mathbf{u}_{i_1}^1 \circ \cdots \circ \mathbf{u}_{i_K}^K$.

### 4.3 CEP

In the context of CEP, the approximation factor $\tilde{f}_{\mathbf{i}}$ is assumed to be factorized with variables, given by

$$\tilde{f}_{\mathbf{i}}(\mathcal{U}, \tau) = \tilde{f}_{\mathbf{i}}(\tau) \prod_{k=1}^{K} \tilde{f}_{\mathbf{i}}^k(\mathbf{u}_{i_k}^k),$$

where the factors have the same form as the prior distribution but with different parameters. Specifically, $\tilde{f}_{\mathbf{i}}(\tau) = \mathrm{Gam}(\tau|a_{\mathbf{i}}, b_{\mathbf{i}})$ and $\tilde{f}_{\mathbf{i}}^k(\mathbf{u}_{i_k}^k) = \mathcal{N}(\mathbf{u}_{i_k}^k | \mathbf{m}_{\mathbf{i}}^k, \mathbf{S}_{\mathbf{i}}^k)$. Consequently, the approximate distribution is given by

$$q(\mathcal{U}, \tau) \propto \mathrm{Gam}(\tau|a_0, b_0) \prod_{k=1}^{K} \prod_{s=1}^{d_k} \mathcal{N}(\mathbf{u}_s^k | \boldsymbol{\beta}_s^k, v \mathbf{I})$$

$$\cdot \prod_{\mathbf{i} \in \mathcal{S}} \tilde{f}_{\mathbf{i}}(\tau) \prod_{k=1}^{K} \tilde{f}_{\mathbf{i}}^k(\mathbf{u}_{i_k}^k).$$

It can be seen that the approximate distribution is factorized over the variables, i.e.,

$$q(\mathcal{U}, \tau) = q(\tau) \prod_{k=1}^{K} \prod_{s=1}^{d_k} q(\mathbf{u}_s^k),$$

where

$$q(\tau) \propto \text{Gam}(\tau|a_0, b_0) \prod_{\mathbf{i} \in \mathcal{S}} \tilde{f}_{\mathbf{i}}(\tau), \tag{28}$$

$$q(\mathbf{u}_s^k) \propto \mathcal{N}(\mathbf{u}_s^k|\boldsymbol{\beta}_s^k, v\mathbf{I}) \prod_{\mathbf{i} \in \mathcal{S}, i_k=s} \tilde{f}_{\mathbf{i}}^k(\mathbf{u}_{i_k}^k).$$

Following the CEP procedure, we construct the calibrating distribution, the tilted distribution, and compute the conditional moments for each variable. The detailed intermediate steps are provided in Appendix F.2. Applying Lemma 4, the optimal factor for each embedding vector is given by $\tilde{f}_{\mathbf{i}}^k(\mathbf{u}_{i_k}^k) = \mathcal{N}(\mathbf{u}_{i_k}^k|\mathbf{m}_{\mathbf{i}}^{k*}, \mathbf{S}_{\mathbf{i}}^{k*})$ with

$$\mathbf{S}_{\mathbf{i}}^{k*} = \left( \langle\tau\rangle \langle \mathbf{z}_{\mathbf{i}}^{\backslash k} \mathbf{z}_{\mathbf{i}}^{\backslash k^T} \rangle \right)^{-1},$$

$$\mathbf{m}_{\mathbf{i}}^{k*} = \mathbf{S}_{\mathbf{i}}^{k*} (y_{\mathbf{i}} \langle\tau\rangle \langle \mathbf{z}_{\mathbf{i}}^{\backslash k} \rangle).$$

Similarly, the optimal message factor for the noise precision is $\tilde{f}_{\mathbf{i}}(\tau) = \text{Gam}(\tau|a_{\mathbf{i}}^*, b_{\mathbf{i}}^*)$ with $a_{\mathbf{i}}^* = \frac{1}{2}$ and $b_{\mathbf{i}}^* = \frac{1}{2}[y_{\mathbf{i}}^2 - 2y_{\mathbf{i}}\langle\mathbf{1}^T\mathbf{z}_{\mathbf{i}}\rangle + \langle(\mathbf{1}^T\mathbf{z}_{\mathbf{i}})^2\rangle]$. It is worth noting that these message factors can be calculated in parallel. After merging all message factors through (28), the optimal variational distribution for $q(\mathbf{u}_s^k)$ is given by $q^*(\mathbf{u}_s^k) = \mathcal{N}(\mathbf{u}_s^k|\boldsymbol{\mu}_s^{k*}, \boldsymbol{\Sigma}_s^{k*})$, where

$$\boldsymbol{\mu}_s^{k*} = \boldsymbol{\Sigma}_s^{k*} \left( \langle\tau\rangle \sum_{\mathbf{i} \in \mathcal{S}, i_k=s} y_{\mathbf{i}} \langle \mathbf{z}_{\mathbf{i}}^{\backslash k} \rangle + v\boldsymbol{\beta}_s^k \right), \tag{29}$$

$$\boldsymbol{\Sigma}_s^{k*} = \left( \langle\tau\rangle \sum_{\mathbf{i} \in \mathcal{S}, i_k=s} \langle \mathbf{z}_{\mathbf{i}}^{\backslash k} \mathbf{z}_{\mathbf{i}}^{\backslash k^T} \rangle + v\mathbf{I} \right)^{-1}.$$

The merged result for $\tau$ is $q^*(\tau) = \text{Gam}(\tau|a^*, b^*)$ with $a^* = a_0 + N/2$ and $b^* = b_0 + \frac{1}{2}\sum_{\mathbf{i} \in \mathcal{S}}[y_{\mathbf{i}}^2 - 2y_{\mathbf{i}}\langle\mathbf{1}^T\mathbf{z}_{\mathbf{i}}\rangle + \langle(\mathbf{1}^T\mathbf{z}_{\mathbf{i}})^2\rangle]$. Comparing (27) and (29), the optimal variational distributions obtained by VMP and CEP are identical, confirming the equivalence established in Theorem 1. These closed-form updates demonstrate promising accuracy and empirically show fast convergence in many real-world applications (Wang & Zhe, 2020).

### 4.4 Streaming VMP

As a direct application of Corollary 2, the VMP-based tensor decomposition algorithm derived in Section 4.2 can be converted to a streaming version without additional derivation effort. In the streaming version, we assume that every time we receive one data point and the current approximation is $q(\mathbf{u}_s^k) = \mathcal{N}(\mathbf{u}_s^k|\boldsymbol{\mu}_s^k, \boldsymbol{\Sigma}_s^k)$ and $q(\tau) = \text{Gam}(\tau|a, b)$. Then based on (23), the posterior update of the streaming version is given by $q^*(\mathbf{u}_s^k) = \hat{p}_{\mathbf{i}}(\mathbf{u}_{i_k}^k|\langle\mathbf{u}_{\mathbf{i}}^{\backslash k}\rangle, \langle\tau\rangle) = \mathcal{N}(\mathbf{u}_s^k|\boldsymbol{\mu}_s^{k*}, \boldsymbol{\Sigma}_s^{k*})$ with

$$\boldsymbol{\Sigma}_s^{k*} = [(\boldsymbol{\Sigma}_s^k)^{-1} + \langle\tau\rangle(\langle\mathbf{z}_{\mathbf{i}}^{\backslash k}\mathbf{z}_{\mathbf{i}}^{\backslash k^T}\rangle)]^{-1}, \tag{30}$$

$$\boldsymbol{\mu}_s^{k*} = \boldsymbol{\Sigma}_s^{k*}[(\boldsymbol{\Sigma}_s^k)^{-1}\boldsymbol{\mu}_s^k + \langle\tau\rangle y_{\mathbf{i}}\langle\mathbf{z}_{\mathbf{i}}^{\backslash k}\rangle],$$

and $q^*(\tau) = \hat{p}_{\mathbf{i}}(\tau|\langle\mathbf{u}_{\mathbf{i}}\rangle) = \text{Gam}(\tau|a^*, b^*)$ with

$$a^* = a + \frac{1}{2},$$

$$b^* = b + \frac{1}{2}[y_{\mathbf{i}}^2 - 2y_{\mathbf{i}}\langle\mathbf{1}^T\mathbf{z}_{\mathbf{i}}\rangle + \langle(\mathbf{1}^T\mathbf{z}_{\mathbf{i}})^2\rangle].$$

Table 1: Reconstruction error (RMSE) for different images using VMP and CEP.

| Image | VMP | CEP |
|---|---|---|
| facade | 0.0184 | 0.0184 |
| car | 0.0365 | 0.0365 |
| airplane | 0.0360 | 0.0360 |
| sailboat | 0.0403 | 0.0403 |

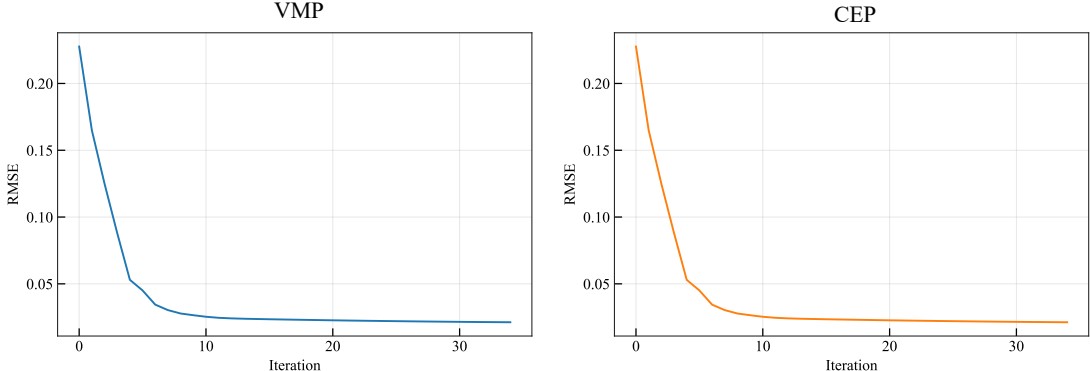

Figure 5: RMSE versus number of iterations for VMP and CEP on the *sailboat* image.

It is important to note that the natural parameters used here are different from those used in CEP (see (30) and (29)) since the calibrating distribution is replaced with the full approximation from the previous iteration. Additionally, this algorithm can be readily extended to the scenario where data arrive in batches. Notably, the resulting streaming updates are identical to those of the probabilistic streaming tensor decomposition (POST) algorithm (Du et al., 2018b), which was originally derived from a different perspective. This illustrates how the established equivalence provides a principled recipe for obtaining streaming variants of VMP-based algorithms.

## 4.5   Numerical Experiments

We evaluate the proposed method on the image inpainting task using four RGB benchmark images[6], each represented as a third-order tensor of size $256 \times 256 \times 3$. The CP tensor rank is set to 40, and all pixel values are normalized to the range $[0, 1]$. A random sampling strategy is used, with an observation rate of 20%. The metric for performance evaluation is the root mean squared error (RMSE). Table 1 reports the average reconstruction error across five independent trials for both VMP and CEP on each image. As expected, the reconstruction results of the two methods are identical, reflecting their equivalent update rules.

Fig. 5 shows a representative example of the convergence behavior for both methods, plotting RMSE against the number of iterations. Both methods exhibit the same convergence trends and reach a low reconstruction error within a few iterations, demonstrating the effectiveness of Bayesian CP decomposition for image completion. Additionally, Fig. 6 presents the visual reconstruction results of VMP and CEP. The recovered images are visually indistinguishable, further confirming the theoretical equivalence of the two methods in practice.

Furthermore, to assess the robustness of both methods beyond the conditions assumed in Theorem 1, we conduct two additional experiments. First, we consider a setting where the group-wise update assumption of CEP is violated. Instead of aggregating messages for each variable group before updating, we adopt a entry-wise update strategy commonly used in standard EP or online implementations, where each update

---

[6]Available: http://sipi.usc.edu/database/database.php

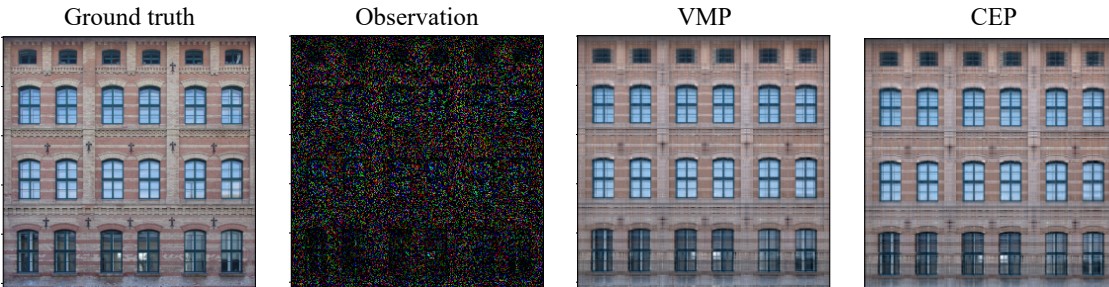

Figure 6: Visual comparison of reconstructed *facade* image using VMP and CEP.

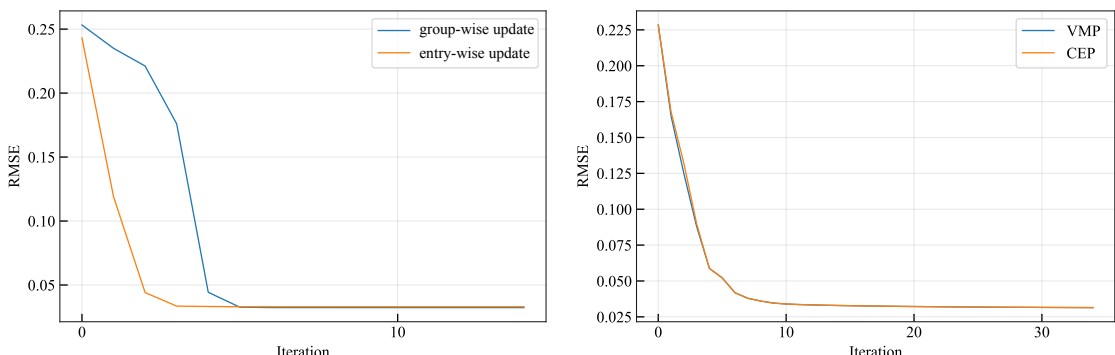

Figure 7: RMSE curves under violation of Theorem 1 assumptions. Left: Convergence behavior of CEP under different update strategies. Right: RMSE curves of VMP and CEP with non-i.i.d. Gaussian noise.

is based on a single data point. As shown in the left panel of Figure 7, CEP still converges reliably despite the lack of a formal convergence guarantee. Moreover, this entry-wise update strategy reduces the number of iterations required, though each iteration incurs higher computational cost.

Second, we test performance of both methods under heteroscedastic noise. Each observed entry is corrupted by Gaussian noise with a data-dependent variance, where $\sigma_i \sim \mathcal{N}(0, 1)$ and the noise is sampled as $\mathcal{N}(0, \sigma_i^2)$. As shown in the right panel of Figure 7, both VMP and CEP maintain identical RMSE levels, demonstrating stability under this non-i.i.d. setting. These results suggest that the convergence and performance consistency between VMP and CEP extend beyond the idealized conditions in the theoretical analysis.

## 5 Discussions and Future Directions

This paper establishes an algorithmic equivalence between VMP and CEP under mild conditions. The key insight is that the variable updates in VMP and CEP are intrinsically merging the messages sent by all the data points, and they share a common objective of approximating the conditional marginal distribution. As a direct consequence, the convergence of CEP is guaranteed under the stated conditions. Furthermore, the equivalence provides a principled recipe for deriving streaming variants of VMP-based algorithms, as demonstrated through the connection between streaming VMP and streaming CEP.

Generally, VMP and CEP are considered distinct classes of ABI methods, each with different properties and performance characteristics. This work, for the first time, establishes a close connection between them under mild conditions, providing new insights into the structure of these algorithms. However, our theoretical analysis is restricted to the conjugate-exponential family of models. It would be interesting to explore the application of these connections in other model families or non-conjugate scenarios. We believe that these explorations will open new avenues for future research on efficient and accurate Bayesian learning algorithms, particularly in the context of streaming and large-scale data.

**Acknowledgments**

The work of Lei Cheng was supported in part by Zhejiang Provincial Natural Science Foundation of China under Grant No. LR26F010004, in part by the National Natural Science Foundation of China under Grant 62371418, and in part by the Key R&D Program of Zhejiang Province 2025C01210. The work of S. Theodoridis was supported in part by the European Union under Horizon Europe (grant No. 101136568 – HERON).

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

## A  Summary of Algorithms

The procedure of the VMP and CEP methods are summarized in **Algorithm 1** and **Algorithm 2**, respectively.

## B  Lemmas

Here we present some useful lemmas to offer a deeper understanding of the exponential family and the KL divergence.

**Lemma 1**(Minka, 2013): *If $p(\boldsymbol{\theta})$ is an arbitrary fixed distribution and $q(\boldsymbol{\theta})$ is in the exponential family, then minimizing the divergence $KL(p\|q)$ with respect to $q$ gives*

$$\mathbb{E}_{q(\boldsymbol{\theta})}[\phi(\boldsymbol{\theta})] = \mathbb{E}_{p(\boldsymbol{\theta})}[\phi(\boldsymbol{\theta})],$$

*where $\phi(\boldsymbol{\theta})$ is the sufficient statistics of $q(\boldsymbol{\theta})$.*

Lemma 1, commonly referred to as moment matching or moment projection, reveals that the KL divergence can be minimized by equating expectations of the sufficient statistics of $q(\boldsymbol{\theta})$ to their expectations with respect to $p(\boldsymbol{\theta})$. It is noteworthy that if $p(\boldsymbol{\theta})$ belongs to the exponential family and shares the same sufficient statistics as $q(\boldsymbol{\theta})$ (i.e., they possess the same distributional form), the moment matching procedure guarantees that their natural parameters become identical. As exponential family distributions are uniquely determined by their sufficient statistics and natural parameters, moment matching leads to the equality of $q(\boldsymbol{\theta})$ and $p(\boldsymbol{\theta})$. Consequently, the KL divergence between the two distributions is reduced to zero.

**Lemma 2**(Bishop, 2006): *Assume $p(\boldsymbol{\theta})$ is a fixed distribution and $q(\boldsymbol{\theta})$ factorizes with respect to variable groups, i.e.,*

$$q(\boldsymbol{\theta}) = \prod_m q(\boldsymbol{\theta}_m),$$

*then minimizing the divergence $KL(p\|q)$ with respect to $q$ gives*

$$q^*(\boldsymbol{\theta}_m) = p(\boldsymbol{\theta}_m), \forall m. \tag{31}$$

Lemma 2 shows that the optimal solution of each factor distribution $q(\boldsymbol{\theta}_m)$ is given by the corresponding marginal distribution of $p(\boldsymbol{\theta})$.

## C  The Delta Approximation Method

The delta approximation method approximates the expectation of a function of a random variable by evaluating the function at the mean of that variable. Specifically, for a function $f(\boldsymbol{\theta})$ and a distribution $q(\boldsymbol{\theta})$, the expectation can be approximated as:

$$\mathbb{E}_{q(\boldsymbol{\theta})}[f(\boldsymbol{\theta})] = \int f(\boldsymbol{\theta})q(\boldsymbol{\theta})\mathrm{d}\boldsymbol{\theta} \approx \int f(\boldsymbol{\theta})\delta(\boldsymbol{\theta} - \mathbf{m})\mathrm{d}\boldsymbol{\theta} = f(\mathbf{m}),$$

where $\delta(\cdot)$ is the Dirac delta function and $\mathbf{m}$ is the mean of $q(\boldsymbol{\theta})$. Since the Dirac delta function can be viewed as the limiting case of a Gaussian distribution with vanishing variance, the delta approximation method can be regarded as a special case of the Laplace approximation (Bishop, 2006).

This approximation can also be interpreted from the perspective of a first-order Taylor expansion. Given a differentiable function $f(\boldsymbol{\theta})$ and a distribution $q(\boldsymbol{\theta})$ with mean $\mathbf{m}$, the first-order Taylor expansion of $f$ around $\mathbf{m}$ yields:

$$\mathbb{E}_{q(\boldsymbol{\theta})}(f(\boldsymbol{\theta})) \approx \mathbb{E}_q\left[f(\mathbf{m}) + (\boldsymbol{\theta} - \mathbf{m})^T \nabla_{\boldsymbol{\theta}} f(\mathbf{m})\right] \approx f(\mathbf{m}),$$

where $\nabla$ is the differential operator. In the context of CEP, a similar approximation can be applied to nested expectations. For example, the outer expectation of a function $h(\boldsymbol{\phi}(\boldsymbol{\theta}_{\backslash m}))$ can be approximated as:

$$\mathbb{E}_{q(\boldsymbol{\theta}_{\backslash m})}[h(\boldsymbol{\phi}(\boldsymbol{\theta}_{\backslash m}))] \approx \mathbb{E}_q\left[h(\mathbb{E}_q(\boldsymbol{\phi}(\boldsymbol{\theta}_{\backslash m}))) + (\boldsymbol{\phi}(\boldsymbol{\theta}_{\backslash m}) - \mathbb{E}_q(\boldsymbol{\phi}(\boldsymbol{\theta}_{\backslash m})))^T \nabla h(\mathbb{E}_q(\boldsymbol{\phi}(\boldsymbol{\theta}_{\backslash m})))\right] \approx h(\mathbb{E}_q(\boldsymbol{\phi}(\boldsymbol{\theta}_{\backslash m}))).$$

Note that in CEP the approximation is applied to the sufficient statistics, rather than the variables themselves.

## D  Graphical Model Derivation

This appendix provides the detailed derivation for the graphical model interpretation presented in Section 3.3. Assume that the variational distribution is fully factorized with respect to the hidden variables. In VMP, the optimized form for each variable is given by

$$\begin{aligned}
\ln q^*(\boldsymbol{\theta}_j) &= \mathbb{E}_{q(\boldsymbol{\theta}_{\backslash j})}[\ln p(\boldsymbol{\theta}, \mathcal{D})] + \mathrm{const}2 \\
&= \langle \ln p(\mathbf{V}) \rangle_{q(\boldsymbol{\theta}_{\backslash j})} + \mathrm{const}2,
\end{aligned} \tag{32}$$

---

**Algorithm 1** Variational Message Passing (VMP)

---

**Input:** joint probability distribution $p(\mathcal{D}, \boldsymbol{\theta})$.

1: Initialise each factor distribution $q(\boldsymbol{\theta}_m)$.
2: **while** not converge **do**
3:     **for** each variable group **do**
4:         Calculate moment of the natural parameters $\mathbb{E}_{q(\boldsymbol{\theta}_{\backslash m})}[\boldsymbol{\eta}_m(\boldsymbol{\theta}_{\backslash m}, \mathcal{D})]$ using the messages sent from other nodes.
5:         Update the factor distribution $q^*(\boldsymbol{\theta}_m)$ via (6).
6:     **end for**
7: **end while**

**Output:** variational distribution $q(\boldsymbol{\theta}) = \prod_m q^*(\boldsymbol{\theta}_m)$.

---

**Algorithm 2** Conditional Expectation Propagation (CEP)

---

**Input:** joint probability distribution $p(\mathcal{D}, \boldsymbol{\theta})$.

1: Initialise each message factor $\tilde{f}_{im}(\boldsymbol{\theta}_m)$.
2: **while** not converge **do**
3:     **for** each variable group **do**
4:         **for** each factor $\tilde{f}_{im}(\boldsymbol{\theta}_m)$ **do**
5:             Calculate the calibrating distribution, $q^{\backslash i}(\boldsymbol{\theta}_m) = q(\boldsymbol{\theta}_m)/\tilde{f}_{im}(\boldsymbol{\theta}_m)$.
6:             Derive a new posterior $q^{\natural}(\boldsymbol{\theta}_m)$ via conditional moment matching (11).
7:             Update the message factor $\tilde{f}_{im}(\boldsymbol{\theta}_m) \propto q^{\natural}(\boldsymbol{\theta}_m)/q^{\backslash i}(\boldsymbol{\theta}_m)$.
8:         **end for**
9:         Merge the message: $q^*(\boldsymbol{\theta}_m) \propto \prod_i \tilde{f}_{im}(\boldsymbol{\theta}_m)$.
10:     **end for**
11: **end while**

**Output:** variational distribution $q(\boldsymbol{\theta}) = \prod_m q^*(\boldsymbol{\theta}_m)$.

---

where $\langle \cdot \rangle_{q(\boldsymbol{\theta}_{\backslash j})}$ denotes the expectation with respect to $q(\boldsymbol{\theta}_{\backslash j})$. Substituting the joint probability distribution into (32) and retaining only the terms that depend on $\boldsymbol{\theta}_j$ (i.e., the Markov blanket of node $j$), we obtain

$$\ln q^*(\boldsymbol{\theta}_j) = \langle \ln p(\boldsymbol{\theta}_j | \mathrm{pa}_j) \rangle_{q(\boldsymbol{\theta}_{\backslash j})} + \sum_{k \in \mathrm{ch}_j} \langle \ln p(\mathbf{v}_k | \mathrm{pa}_k) \rangle_{q(\boldsymbol{\theta}_{\backslash j})} + \mathrm{const5}, \tag{33}$$

where $\mathrm{ch}_j$ denotes the index set that corresponds to the children of node $j$. The parent node of $\mathbf{v}_k$ includes the node $j$ and the co-parents $\mathrm{cp}_j$.

In a conjugate-exponential model, we have

$$\ln p(\boldsymbol{\theta}_j | \mathrm{pa}_j) = \boldsymbol{\eta}_j(\mathrm{pa}_j)^T \boldsymbol{\phi}_j(\theta_j) + Z_j(\mathrm{pa}_j) + \ln h_j(\boldsymbol{\theta}_j), \tag{34}$$

and

$$\begin{aligned}\ln p(\mathbf{v}_k | \mathrm{pa}_k) &= \boldsymbol{\eta}_k(\boldsymbol{\theta}_j, \mathrm{cp}_j)^T \boldsymbol{\phi}_k(\mathbf{v}_k) + Z_k(\boldsymbol{\theta}_j, \mathrm{cp}_j) + \ln h_k(\mathbf{v}_k) \\ &= \boldsymbol{\eta}_{kj}(\mathbf{v}_k, \mathrm{cp}_j)^T \boldsymbol{\phi}_j(\boldsymbol{\theta}_j) + \lambda(\mathbf{v}_k, \mathrm{cp}_j),\end{aligned} \tag{35}$$

where $\lambda$ is a function that contains the terms irrelevant to $\boldsymbol{\eta}_{kj}$ and $\boldsymbol{\phi}_j(\boldsymbol{\theta}_j)$. The second equation holds due to the conjugacy property. Substituting (34) and (35) into (33) gives

$$\ln q^*(\boldsymbol{\theta}_j) = \left[ \langle \boldsymbol{\eta}_j(\mathrm{pa}_j) \rangle_{q(\boldsymbol{\theta}_{\backslash j})} + \sum_{k \in \mathrm{ch}_j} \langle \boldsymbol{\eta}_{kj}(\mathbf{v}_k, \mathrm{cp}_j) \rangle_{q(\boldsymbol{\theta}_{\backslash j})} \right]^T \boldsymbol{\phi}_j(\boldsymbol{\theta}_j) + \ln h_j(\boldsymbol{\theta}_j) + \mathrm{const6}.$$

It follows that the optimal variational distribution $q^*(\boldsymbol{\theta}_j)$ is also an exponential family distribution with the same form as $p(\boldsymbol{\theta}_j|\text{pa}_j)$, of which the natural parameter is

$$\boldsymbol{\eta}_j^* = \langle \boldsymbol{\eta}_j(\text{pa}_j) \rangle + \sum_{k \in \text{ch}_j} \langle \boldsymbol{\eta}_{kj}(\mathbf{v}_k, \text{cp}_j) \rangle,$$

where the expectations are with respect to $q(\boldsymbol{\theta}_{\backslash j})$. As the probabilistic model is conjugate-exponential, we can reparameterise these functions in terms of the expected sufficient statistics, leading to (24).

For the CEP side, the optimal variational distribution can be expressed as

$$\ln q^*(\boldsymbol{\theta}_j) = \ln p(\boldsymbol{\theta}_j|\text{pa}_j) + \sum_{k=1}^{N} \ln \tilde{f}_k(\boldsymbol{\theta}_j),$$

where $\ln p(\boldsymbol{\theta}_j|\text{pa}_j)$ is given by (34). From Lemma 5, $\ln \tilde{f}_k(\boldsymbol{\theta}_j) = \mathbb{E}_{q(\boldsymbol{\theta}_{\backslash j})}[\ln f_k(\boldsymbol{\theta}_j, \boldsymbol{\theta}_{\backslash j})]$. Using (35), we have $\ln f_k(\boldsymbol{\theta}_j, \boldsymbol{\theta}_{\backslash j}) = \boldsymbol{\eta}_{kj}(\mathbf{v}_k, \text{cp}_j)^T \boldsymbol{\phi}_j(\boldsymbol{\theta}_j) + \lambda(\mathbf{v}_k, \text{cp}_j)$. Taking the expectation and reparameterizing yields the expression shown in the main text, confirming that the resulting distribution shares the same natural parameters as in (24).

# E    Proofs

## E.1    Proof of Lemma 3

To prove Lemma 3, we rewrite the KL divergence as

$$\text{KL}(p\|q) = \int p(\boldsymbol{\theta}) \ln \frac{p(\boldsymbol{\theta})}{q(\boldsymbol{\theta})} d\boldsymbol{\theta}$$

$$= H[p(\boldsymbol{\theta})] - \int p(\boldsymbol{\theta}) \ln q(\boldsymbol{\theta}) d\boldsymbol{\theta},$$

where $H[\cdot]$ is the entropy. Since the entropy is a constant, minimizing $\text{KL}(p\|q)$ is equivalent to maximizing $\mathcal{L}(q) = \int p(\boldsymbol{\theta}) \ln q(\boldsymbol{\theta}) d\boldsymbol{\theta}$. Exploiting the factorized property, it can be further decomposed as

$$\mathcal{L}(q) = \int p(\boldsymbol{\theta}) \sum_m \ln q(\boldsymbol{\theta}_m) d\boldsymbol{\theta}$$

$$= \sum_m \int p(\boldsymbol{\theta}) \ln q(\boldsymbol{\theta}_m) d\boldsymbol{\theta}$$

$$= \sum_m \int \left( \int p(\boldsymbol{\theta}) d\boldsymbol{\theta}_{\backslash m} \right) \ln q(\boldsymbol{\theta}_m) d\boldsymbol{\theta}_m$$

$$= \sum_m \int p(\boldsymbol{\theta}_m) \ln q(\boldsymbol{\theta}_m) d\boldsymbol{\theta}_m$$

$$= \sum_m L_m(q(\boldsymbol{\theta}_m)),$$

where we denote $\int p(\boldsymbol{\theta}_m) \ln q(\boldsymbol{\theta}_m) d\boldsymbol{\theta}_m$ as $L_m(q(\boldsymbol{\theta}_m))$. Since the variable groups are mutually independent, maximizing $\mathcal{L}(q)$ with respect to $q(\boldsymbol{\theta})$ is equivalent to maximizing each $L_m$ with respect to $q(\boldsymbol{\theta}_m)$. For each variable group, the optimum is given by

$$\max_{q(\boldsymbol{\theta}_m)} L_m(q(\boldsymbol{\theta}_m)) = \max_{q(\boldsymbol{\theta}_m)} \int p(\boldsymbol{\theta}_m) \ln q(\boldsymbol{\theta}_m) d\boldsymbol{\theta}_m$$

$$= \min_{q(\boldsymbol{\theta}_m)} - \int p(\boldsymbol{\theta}_m) \ln q(\boldsymbol{\theta}_m) d\boldsymbol{\theta}_m$$

$$= \min_{q(\boldsymbol{\theta}_m)} H[p(\boldsymbol{\theta}_m)] - \int p(\boldsymbol{\theta}_m) \ln q(\boldsymbol{\theta}_m) d\boldsymbol{\theta}_m$$

$$= \min_{q(\boldsymbol{\theta}_m)} \text{KL}(p(\boldsymbol{\theta}_m)\|q(\boldsymbol{\theta}_m)),$$

where the third equation holds because the entropy of the marginal distribution $H[p(\boldsymbol{\theta}_m)]$ is irrelevant to $q(\boldsymbol{\theta}_m)$. Using Lemma 1, the optimal solution is achieved by the moment matching

$$\mathbb{E}_{q(\boldsymbol{\theta}_m)}[\phi(\boldsymbol{\theta}_m)] = \mathbb{E}_{p(\boldsymbol{\theta}_m)}[\phi(\boldsymbol{\theta}_m)].$$

Additionally, the left-hand side of (12) can be expressed as

$$\begin{aligned}
\mathbb{E}_{q(\boldsymbol{\theta})}[\phi(\boldsymbol{\theta}_m)] &= \int q(\boldsymbol{\theta})\phi(\boldsymbol{\theta}_m)d\boldsymbol{\theta} \\
&= \int \left[\int q(\boldsymbol{\theta}_m, \boldsymbol{\theta}_{\backslash m})d\boldsymbol{\theta}_{\backslash m}\right]\phi(\boldsymbol{\theta}_m)d\boldsymbol{\theta}_m \\
&= \int q(\boldsymbol{\theta}_m)\phi(\boldsymbol{\theta}_m)d\boldsymbol{\theta}_m \\
&= \mathbb{E}_{q(\boldsymbol{\theta}_m)}[\phi(\boldsymbol{\theta}_m)].
\end{aligned}$$

Similarly, the right-hand side of (12) can be expressed as $\mathbb{E}_{p(\boldsymbol{\theta})}[\phi(\boldsymbol{\theta}_m)] = \mathbb{E}_{p(\boldsymbol{\theta}_m)}[\phi(\boldsymbol{\theta}_m)]$. Thus we have

$$\mathbb{E}_{q(\boldsymbol{\theta})}[\phi(\boldsymbol{\theta}_m)] = \mathbb{E}_{q(\boldsymbol{\theta}_m)}[\phi(\boldsymbol{\theta}_m)] = \mathbb{E}_{p(\boldsymbol{\theta}_m)}[\phi(\boldsymbol{\theta}_m)] = \mathbb{E}_{p(\boldsymbol{\theta})}[\phi(\boldsymbol{\theta}_m)],$$

which completes the proof. ∎

### E.2 Proof of Lemma 4

From (13) and Lemma 3, it can be seen that the calculation of $q^{\natural}(\boldsymbol{\theta}_m)$ in CEP is essentially solving the following problem

$$\begin{aligned}
\min_{q(\boldsymbol{\theta}_m)} \ &\mathrm{KL}(\hat{p}_i(\boldsymbol{\theta})\|q(\boldsymbol{\theta})) \\
\text{s.t. } \ &q(\boldsymbol{\theta}) = \prod_m q(\boldsymbol{\theta}_m),
\end{aligned}$$

where $q(\boldsymbol{\theta}_m)$ belongs to the exponential family. If the $\hat{p}_i(\boldsymbol{\theta}_m)$ is also in the exponential family and has the same form as $q^{\natural}(\boldsymbol{\theta}_m)$, then the moment matching leads to

$$q^{\natural}(\boldsymbol{\theta}_m) = \hat{p}_i(\boldsymbol{\theta}_m),$$

where the marginal posterior can be further written as

$$\begin{aligned}
\hat{p}_i(\boldsymbol{\theta}_m) &= \int \hat{p}_i(\boldsymbol{\theta}_m, \boldsymbol{\theta}_{\backslash m})d\boldsymbol{\theta}_{\backslash m} \\
&= \int \hat{p}_i(\boldsymbol{\theta}_{\backslash m})\hat{p}_i(\boldsymbol{\theta}_m|\boldsymbol{\theta}_{\backslash m})d\boldsymbol{\theta}_{\backslash m} \\
&= \mathbb{E}_{\hat{p}_i(\boldsymbol{\theta}_{\backslash m})}[\hat{p}_i(\boldsymbol{\theta}_m|\boldsymbol{\theta}_{\backslash m})].
\end{aligned}$$

In CEP, two approximations are made to derive an analytical form of the update. The first is to use $q(\boldsymbol{\theta}_{\backslash m})$ as a surrogate for $\hat{p}_i(\boldsymbol{\theta}_{\backslash m})$. The second is to use the delta approximation method to approximate the expectation of the conditional distribution. Based on these approximations, the optimal approximate posterior $q^{\natural}(\boldsymbol{\theta}_m)$ can be expressed as

$$\begin{aligned}
q^{\natural}(\boldsymbol{\theta}_m) &= \mathbb{E}_{\hat{p}_i(\boldsymbol{\theta}_{\backslash m})}[\hat{p}_i(\boldsymbol{\theta}_m|\boldsymbol{\theta}_{\backslash m})] \\
&\approx \mathbb{E}_{q(\boldsymbol{\theta}_{\backslash m})}[\hat{p}_i(\boldsymbol{\theta}_m|\boldsymbol{\theta}_{\backslash m})] \\
&\approx \hat{p}_i(\boldsymbol{\theta}_m|\mathbb{E}_q[\phi(\boldsymbol{\theta}_{\backslash m})]).
\end{aligned}$$

Note that the delta approximation is made on the sufficient statistics, rather than the random variables themselves.

Generally, $\hat{p}_i(\boldsymbol{\theta}_m)$ is not in the exponential family, so the moment matching step is used to minimize the KL divergence. However, in a conjugate-exponential model, each complete conditional, including $\hat{p}_i(\boldsymbol{\theta}_m|\boldsymbol{\theta}_{\backslash m})$, is in the exponential family. Additionally, $\hat{p}_i(\boldsymbol{\theta}_m|\boldsymbol{\theta}_{\backslash m})$ shares the same sufficient statistics as $q^{\natural}(\boldsymbol{\theta}_m)$ due to the conjugacy property. As a result, $\hat{p}_i(\boldsymbol{\theta}_m|\mathbb{E}_q[\boldsymbol{\phi}(\boldsymbol{\theta}_{\backslash m})])$ is used as a surrogate for $q^{\natural}(\boldsymbol{\theta}_m)$ in CEP. Thus the update of $\tilde{f}_{im}(\boldsymbol{\theta}_m)$ can be expressed as

$$\tilde{f}_{im}(\boldsymbol{\theta}_m) \propto \frac{\hat{p}_i(\boldsymbol{\theta}_m|\mathbb{E}_q[\boldsymbol{\phi}(\boldsymbol{\theta}_{\backslash m})])}{q^{\backslash i}(\boldsymbol{\theta}_m)},$$

which completes the proof. ∎

### E.3 Proof of Corollary 1

It has been established in Winn et al. (2005); Minka (2005) that VMP updates are guaranteed to converge to a local minimum of the KL divergence under the conditions stated in Theorem 1. Since CEP follows the same update equations as VMP under these conditions, its convergence property directly follows.

### E.4 Proof of Corollary 2

We first derive the streaming CEP update by applying the two approximations in Corollary 2 to ADF. In ADF, the optimal variational distribution in each iteration can be expressed as

$$q^*(\boldsymbol{\theta}_m) = \hat{p}_i(\boldsymbol{\theta}_m) = \mathbb{E}_{\hat{p}_i(\boldsymbol{\theta}_{\backslash m})}[\hat{p}_i(\boldsymbol{\theta}_m|\boldsymbol{\theta}_{\backslash m})].$$

With the two conditions in Corollary 2, the optimal distribution can be reformulated as

$$\begin{aligned} q^*(\boldsymbol{\theta}_m) &= \mathbb{E}_{\hat{p}_i(\boldsymbol{\theta}_{\backslash m})}[\hat{p}_i(\boldsymbol{\theta}_m|\boldsymbol{\theta}_{\backslash m})] \\ &\approx \mathbb{E}_{q(\boldsymbol{\theta}_{\backslash m})}[\hat{p}_i(\boldsymbol{\theta}_m|\boldsymbol{\theta}_{\backslash m})] \\ &\approx \hat{p}_i(\boldsymbol{\theta}_m|\mathbb{E}_q[\boldsymbol{\phi}(\boldsymbol{\theta}_{\backslash m})]). \end{aligned}$$

Similarly, the optimal distribution in streaming VMP is given by

$$\begin{aligned} \ln q^*(\boldsymbol{\theta}_m) &= \mathbb{E}_{q(\boldsymbol{\theta}_{\backslash m})}[\ln \hat{p}_i(\boldsymbol{\theta}_m|\boldsymbol{\theta}_{\backslash m})] \\ &= \ln \hat{p}_i(\boldsymbol{\theta}_m|\mathbb{E}_q[\boldsymbol{\phi}(\boldsymbol{\theta}_{\backslash m})]), \end{aligned}$$

which is the same as the streaming CEP update derived above. This completes the proof.

## F Derivations for Tensor Decomposition

### F.1 VMP Derivation

In the Bayesian tensor decomposition problem, the unknown parameter set $\boldsymbol{\theta}$ consists of the latent factor matrices $\mathcal{U}$ and hyperparameter $\tau$. The optimal variational distribution for each $\boldsymbol{\theta}_m$ is given by

$$\ln q^*(\boldsymbol{\theta}_m) = \mathbb{E}_{q(\boldsymbol{\theta}_{\backslash m})}[\ln p(\boldsymbol{\theta}, \mathcal{D})] + \text{const.} \tag{36}$$

From (26), the logarithm of the joint density function $\ln p(\boldsymbol{\theta}, \mathcal{D})$ can be expressed as

$$\begin{aligned} \ln p(\boldsymbol{\theta}, \mathcal{D}) &= \ln p(\{y_{\mathbf{i}}\}_{\mathbf{i} \in \mathcal{S}}, \mathcal{U}, \tau) \\ &= \frac{N}{2}\ln\tau - \sum_{\mathbf{i}\in\mathcal{S}}\frac{\tau}{2}[y_{\mathbf{i}} - \mathbf{1}^T(\mathbf{u}_{i_1}^1 \circ \cdots \circ \mathbf{u}_{i_K}^K)]^2 \\ &\quad - \sum_{k=1}^{K}\sum_{s=1}^{d_k}\frac{v}{2}(\mathbf{u}_s^k - \boldsymbol{\beta}_s^k)^T(\mathbf{u}_s^k - \boldsymbol{\beta}_s^k) \\ &\quad (a_0 - 1)\ln\tau - b_0\tau + \text{const.} \end{aligned} \tag{37}$$

By substituting (37) into (36), we obtain $q^*(\mathbf{u}_s^k)$:

$$
\begin{aligned}
\ln q^*(\mathbf{u}_s^k) &= \mathbb{E}_q\{-\frac{\tau}{2}\sum_{\mathbf{i}\in\mathcal{S},i_k=s}[y_\mathbf{i} - \mathbf{1}^T(\mathbf{u}_{i_1}^1 \circ \cdots \circ \mathbf{u}_{i_K}^K)]^2 \\
&\quad -\frac{v}{2}(\mathbf{u}_s^k - \boldsymbol{\beta}_s^k)^T(\mathbf{u}_s^k - \boldsymbol{\beta}_s^k)\} \\
&= \mathbb{E}_q\{-\frac{\tau}{2}\sum_{\mathbf{i}\in\mathcal{S},i_k=s}\left[y_\mathbf{i}^2 - 2y_\mathbf{i}(\mathbf{u}_s^k)^T\mathbf{z}_\mathbf{i}^{\backslash k} + (\mathbf{u}_s^k)^T\mathbf{z}_\mathbf{i}^{\backslash k}\mathbf{z}_\mathbf{i}^{\backslash k^T}\mathbf{u}_s^k\right] \\
&\quad -\frac{v}{2}\left[(\mathbf{u}_s^k)^T\mathbf{u}_s^k - 2(\mathbf{u}_s^k)^T\boldsymbol{\beta}_s^k + (\boldsymbol{\beta}_s^k)^T\boldsymbol{\beta}_s^k\right]\} \\
&= -\frac{1}{2}(\mathbf{u}_s^k)^T\left[\langle\tau\rangle\sum_{\mathbf{i}\in\mathcal{S},i_k=s}\langle\mathbf{z}_\mathbf{i}^{\backslash k}\mathbf{z}_\mathbf{i}^{\backslash k^T}\rangle + v\mathbf{I}\right]\mathbf{u}_s^k \\
&\quad + (\mathbf{u}_s^k)^T\left[\langle\tau\rangle\sum_{\mathbf{i}\in\mathcal{S},i_k=s}y_\mathbf{i}\langle\mathbf{z}_\mathbf{i}^{\backslash k}\rangle + v\boldsymbol{\beta}_s^k\right].
\end{aligned} \tag{38}
$$

We can see from (38) that $\mathbf{u}_s^k$ follows a Gaussian distribution $q^*(\mathbf{u}_s^k) = \mathcal{N}(\mathbf{u}_s^k|\boldsymbol{\mu}_s^{k*}, \boldsymbol{\Sigma}_s^{k*})$, of which the mean and covariance are given by

$$
\boldsymbol{\mu}_s^{k*} = \boldsymbol{\Sigma}_s^{k*}\left(\langle\tau\rangle\sum_{\mathbf{i}\in\mathcal{S},i_k=s}y_\mathbf{i}\langle\mathbf{z}_\mathbf{i}^{\backslash k}\rangle + v\boldsymbol{\beta}_s^k\right),
$$

$$
\boldsymbol{\Sigma}_s^{k*} = \left(\langle\tau\rangle\sum_{\mathbf{i}\in\mathcal{S},i_k=s}\langle\mathbf{z}_\mathbf{i}^{\backslash k}\mathbf{z}_\mathbf{i}^{\backslash k^T}\rangle + v\mathbf{I}\right)^{-1}.
$$

The expression of $q^*(\tau)$ can be found as

$$
\begin{aligned}
\ln q^*(\tau) &= \mathbb{E}_q\{\frac{N}{2}\ln\tau - \sum_{\mathbf{i}\in\mathcal{S}}\frac{\tau}{2}[y_\mathbf{i} - \mathbf{1}^T(\mathbf{u}_{i_1}^1 \circ \cdots \circ \mathbf{u}_{i_K}^K)]^2 \\
&\quad (a_0 - 1)\ln\tau - b_0\tau\},
\end{aligned}
$$

which is a Gamma distribution $q^*(\tau) = \mathrm{Gam}(\tau|a^*, b^*)$ with $a^*$ and $b^*$ given by

$$
a^* = a_0 + \frac{N}{2},
$$

$$
b^* = b_0 + \frac{1}{2}\sum_{\mathbf{i}\in\mathcal{S}}[y_\mathbf{i}^2 - 2y_\mathbf{i}\langle\mathbf{1}^T\mathbf{z}_\mathbf{i}\rangle + \langle(\mathbf{1}^T\mathbf{z}_\mathbf{i})^2\rangle].
$$

### F.2 Derivation of CEP

To update the variational distribution $q(\mathbf{u}_s^k)$ in CEP, we first obtain the calibrating distribution

$$
q^{\backslash\mathbf{i}}(\mathcal{U},\tau) \propto \frac{q(\mathcal{U},\tau)}{\tilde{f}_\mathbf{i}(\tau)\prod_{k=1}^K \tilde{f}_\mathbf{i}^k(\mathbf{u}_{i_k}^k)}.
$$

Next, we construct the tilted distribution as

$$
\hat{p}_\mathbf{i}(\mathcal{U},\tau) \propto q^{\backslash\mathbf{i}}(\mathcal{U},\tau)\mathcal{N}(y_\mathbf{i}|\mathbf{1}^T(\mathbf{u}_{i_1}^1 \circ \cdots \circ \mathbf{u}_{i_K}^K), \tau^{-1}).
$$

Since only the moments for $\tau$ and the embedding vectors associated with entry $\mathbf{i}$ are needed, we focus on the marginal tilted distribution

$$
\hat{p}_\mathbf{i}(\mathbf{u}_\mathbf{i},\tau) \propto q^{\backslash\mathbf{i}}(\tau)\prod_{k=1}^K q^{\backslash\mathbf{i}}(\mathbf{u}_{i_k}^k)\mathcal{N}(y_\mathbf{i}|\mathbf{1}^T(\mathbf{u}_{i_1}^1 \circ \cdots \circ \mathbf{u}_{i_K}^K), \tau^{-1}),
$$

where $q^{\backslash \mathbf{i}}(\tau) = \text{Gam}(\tau | a^{\backslash \mathbf{i}}, b^{\backslash \mathbf{i}})$ and $q^{\backslash \mathbf{i}}(\mathbf{u}_{i_k}^k) = \mathcal{N}(\mathbf{u}_{i_k}^k | \mathbf{m}_{i_k}^k, \mathbf{S}_{i_k}^k)$, with

$$a^{\backslash \mathbf{i}} = a_0 + \sum_{\mathbf{j} \in \mathcal{S}, \mathbf{j} \neq \mathbf{i}} a_{\mathbf{j}} - N + 1, \quad b^{\backslash \mathbf{i}} = b_0 + \sum_{\mathbf{j} \in \mathcal{S}, \mathbf{j} \neq \mathbf{i}} b_{\mathbf{j}},$$

$$\mathbf{S}_{i_k}^k = \left( \sum_{\mathbf{j} \in \mathcal{S}, \mathbf{j} \neq \mathbf{i}, j_k = i_k} (\mathbf{S}_{\mathbf{j}}^k)^{-1} + v\mathbf{I} \right)^{-1},$$

$$\mathbf{m}_{i_k}^k = \mathbf{S}_{i_k}^k \left( \sum_{\mathbf{j} \in \mathcal{S}, \mathbf{j} \neq \mathbf{i}, j_k = i_k} (\mathbf{S}_{\mathbf{j}}^k)^{-1} \mathbf{m}_{\mathbf{j}}^k + v\beta_{i_k}^k \right).$$

The conditional tilted distribution for $\mathbf{u}_{i_k}^k$ given $\tau$ and $\mathbf{u}_{\mathbf{i}}^{\backslash k}$ fixed is

$$\hat{p}_{\mathbf{i}}(\mathbf{u}_{i_k}^k | \mathbf{u}_{\mathbf{i}}^{\backslash k}, \tau) \propto \mathcal{N}(\mathbf{u}_{i_k}^k | \mathbf{m}_{i_k}^k, \mathbf{S}_{i_k}^k) \mathcal{N}(y_{\mathbf{i}} | \mathbf{1}^T (\mathbf{u}_{i_1}^1 \circ \cdots \circ \mathbf{u}_{i_K}^K), \tau^{-1}),$$

which is a Gaussian distribution with covariance and mean given by

$$\text{cov}(\mathbf{u}_{i_k}^k | \mathbf{u}_{\mathbf{i}}^{\backslash k}, \tau) = \left[ (\mathbf{S}_{i_k}^k)^{-1} + \tau (\mathbf{z}_{\mathbf{i}}^{\backslash k} \mathbf{z}_{\mathbf{i}}^{\backslash k T}) \right]^{-1},$$

$$\mathbb{E}(\mathbf{u}_{i_k}^k | \mathbf{u}_{\mathbf{i}}^{\backslash k}, \tau) = \text{cov}(\mathbf{u}_{i_k}^k | \mathbf{u}_{\mathbf{i}}^{\backslash k}, \tau) \left[ (\mathbf{S}_{i_k}^k)^{-1} \mathbf{m}_{i_k}^k + \tau y_{\mathbf{i}} \mathbf{z}_{\mathbf{i}}^{\backslash k} \right].$$

For the noise precision $\tau$, the conditional tilted distribution is $\hat{p}_{\mathbf{i}}(\tau | \mathbf{u}_{\mathbf{i}}) = \text{Gam}(\tau | \hat{a}, \hat{b})$ with $\hat{a} = a^{\backslash \mathbf{i}} + \frac{1}{2}$ and $\hat{b} = b^{\backslash \mathbf{i}} + \frac{1}{2}[y_{\mathbf{i}} - \mathbf{1}^T \mathbf{z}_{\mathbf{i}}]^2$. Applying Lemma 4 yields the optimal factors presented in Section 4.3.

