# OpenReview forum: "When VMP Meets CEP: An Algorithmic Equivalence Under Mild Conditions"
_TMLR — Accepted by TMLR_

### Review · Reviewer_22CT · 2026-01-30

**Summary Of Contributions:**

The paper studies connections between two approximate Bayesian inference methods: variational message pasing and conditional expectation propagation, in terms of their performance and convergence properties.

**Audience:**

Yes

**Audience Explanation:**

The paper is of interest to the theoretical ML community.

**Broader Impact Concerns:**

N.A.

**Claims And Evidence:**

Yes

**Claims Explanation:**

I believe the technical flaw that was pointed out by a reviewer in the original submission of this work has been resolved.

- Equivalence between VMP and CEP is now explicitly conditional (Theorem 1)
- The delta approximation is elevated from an implicit step to a formal assumption (Appendix C)
- Lemma 5 is reframed as a sufficient condition for equivalence (Eq (15))

The revised paper does not seem to have changed its core mathematical result, but it restricts and correctly conditions it.

**Requested Changes:**

I am happy with the manuscript as is.

---

> ### Author Response · Authors · 2026-04-15
>
> We thank the reviewer for the positive feedback and a clear summary of the theoretical results. Based on the feedback from the other reviewers, we have also made revisions to the presentation, including refining the title and clarifying certain notation and terminology. We believe the manuscript with these improvements is now clearer and more precise.

---

### Review · Reviewer_GneH · 2026-02-28

**Summary Of Contributions:**

The authors show an equivalence between variational message passing (VMP) and conditional expectation propagation (CEP) under certain conditions. I believe this is a novel result that will be of interest to some of the TMLR audience.  The authors also make some claims about the contributions of the paper that I believe are unjustified. Another weakness of the paper in my opinion is that there is a large amount of real estate dedicated to filler content that seems unnecessary and largely unrelated to the main result.  There are also a number of presentational issues, but I believe these easily be easily fixed.

**Audience:**

Yes

**Audience Explanation:**

The equivalence of VMP and CEP under the stated conditions is novel and will be of interest to those interested in such methods.

**Broader Impact Concerns:**

No broader impact concerns.

**Claims And Evidence:**

No

**Claims Explanation:**

The main theoretical result is novel and I believe backed up by the author's proofs.  However, the paper also makes some related claims in the abstract (and elsewhere) that I believe are unjustified. Specifically:

1. _"This newly unveiled connection on only enhances our understanding of the performance and convergence properties of VMP and CEP, but it also facilitates the cross-fertilization of their respective strengths"_. The theoretical result does not, as far as I can see, enhance our understanding of the performance/convergence of VMP in anyway, nor is it leveraged to demonstrate any improvement over VMP.

2. _"we ... demonstrate how this connection facilitates the construction of streaming VMP"_. I am uncomfortable with this aspect of the paper, as streaming/online VMP is well studied, and it is not clear there is any contribution here beyond a simple corollary of the main result -- namely that a streaming version of CEP is also the same as streaming VMP.

3. _"... shedding new light on the understanding and development of more advanced ABI techniques"_. I don't believe this is justified - what are these more advanced ABI techniques? I don't think any new techniques are discussed in the paper.

I believe the main result alone is enough, so I think the paper would be better served by removing these claims completely.

**Requested Changes:**

To vote for acceptance I would want to see the claimed contributions mentioned above either removed completely (from the abstract and throughout), or justified in concrete terms.  There are also many problems with the presentation, which need to be addressed in my opinion.

1. Figure 1 is misleading, because it implies that ADF and the streaming version of CEP are the same thing.  The equivalence is between streaming VI and streaming CEP, which is distinct from ADF.  I think this needs to be corrected, e.g. by separately adding streaming CEP (or some other name), with a line showing that it is obtained from ADF using additional CEP approximations.
2. The claim (p2) that the streaming VMP algorithm (Broderick et al. 2013) _"is not straightforward and demands additional effort"_ is not justified.  The paper's result shows that streaming CEP and VMP are the same thing (under certain settings), so it can't also claim to be more straightforward and demand less effort.  I think the paper should just drop the claim of novelty on the streaming VMP altogether.
3. A equivalence between power EP and VI in the limiting case (of the power parameter) has been shown by Bui et al. 2018. This seems relevant and should be discussed somewhere.
4. _"The connection can also provie some insights into the convergence of the standard EP"_ (p3) - I don't think this is true, but if it is please state explicitly what those insights are, otherwise remove this claim.
5.  _"facilitates the seamless construction of an online or distributed variant of VMP"_ (p3) - again, online VMP is already well known, so please take out this claim.
6. On p5 the second line of equation $\\mathcal{L}(q) = $ is missing a summation
7. On p6 _"builds on ADF"_ -> _"generalizes ADF"_
8. On p6 _"corresponds to the likelihood"_ -> _"may correspond to the likelihood"_
9. On p8 (and later), it is confusing that $\\tilde f_i(\\theta)$ and $\\tilde f_i(\\theta_m)$ are different functions. Later in the paper (on p12) the authors use the notation $\\tilde f_im(\\theta_m)$ to make the distinction clear -- it would be better to use that notation consistently throughout.
10. Similarly the statistics function $\\phi(\\theta_m)$ is group-specific (can vary by m), so it would be clearer to also subscript this in my opinion.
11. p8 _"can be represented with a quadrature formula"_ -> "can be computed using quadrature methods"
12. p8 "the conditional moment $E_{\\hat p_i(\\theta_m\\mid\\theta_{\\textbackslash m})}[\\phi(\\theta_m)]$ is a function of the sufficient statistics of $\\theta_{\\textbackslash m}$".  This is not necessarily true, it requires additional assumptions.  It is true if $\theta_m$ is fully recoverable from $\\phi(\\theta_m)$ (not always the case), or if the density $p_i(\theta_m\mid\theta_{\\textbackslash m})$ depends on $\\theta_{\\textbackslash m}$ only through the statistics function $\phi(.)$ -- but in this section we have not made any assumptions about the form of the model.  Can you clarify what additional assumptions are being made here?  I believe this was also left unspecified in the original CEP paper, but it would be good to provide clarification here.
13. p8 $\phi(.)$ is used to denote the statistic functions of both the approximating family, as well as the statistic functions of the conditional exponential family of the model.  Are we assuming these functions are always the same? Please make this explicit if so, otherwise it would be clearer to use alternative notation if they are / can be distinct.
14. p10 "where the tilted distribution $q^{\\textbackslash i}(\\theta)$ is defined in (8)" -- this should say cavity/calibrated distribution, and it is not defined in (8).
15. In (14), the notation $\\hat{p}\_i(\\theta\_m\\mid\\mathbb{E}\_q[\\phi(\\theta\_{\\textbackslash m})])$ is not clear. I believe you are using this as shorthand for the density with the same form as $\\hat{p}\_i(\\theta\_m\\mid\\theta_{\\textbackslash m})$, but where the statistic functions $\\phi(\\theta\_{\\textbackslash m})])$ are replaced by their expectations under $q$. Its form is implicitly given in the last two lines of (20) -- this should be used to define it explicitly earlier on, before it is used.
16. p12 "using $q(\\theta\_m)$ as a surrogate of $p(\\theta\_m)$", I think these should be $q(\\theta\_{\\textbackslash m})$  and $p(\\theta\_{\\textbackslash m}\\mid\\mathcal{D})$
17. The authors switch from using $\mathbb{E}[.]$ to using $\langle . \rangle$ to denote expectations halfway through (25) and for the remainder of that section, for seemingly no reason. It gives the impression that separate pieces of work have been stitched together with no attempt to ensure consistency. I would recommend using one notation throughout.

I have read the paper several times now and each read through I find multiple new errors. I would really recommend the authors carefully proofread the submission to check for errors. Also as a general comment, I feel almost everything after page 12 is just filler content and does not really add anything to the paper -- I won't make drastic changes here a condition of my acceptance, but I would sincerely recommend the authors consider drastically cutting some of this content, as I think the paper would be better for it.

---

> ### Author Response · Authors · 2026-04-15
>
> We thank the reviewer for the detailed and constructive feedback, and for acknowledging that the main theoretical result is novel and backed up by the proofs. We have carefully addressed each of the reviewer's concerns. Below, we summarize the main issues raised and provide point-by-point responses:
>
> - Unjustified claims in the abstract and introduction;
> - Misleading Figure 1 regarding ADF and streaming CEP;
> - Missing discussion of Bui et al. (2017);
> - Presentation and notation issues (items 6--17);
> - Proofreading and content after page 12.
>
> *C1. (Claims Explanation) The paper makes claims that are unjustified: (A) "This newly unveiled connection not only enhances our understanding of the performance and convergence properties of VMP and CEP, but it also facilitates the cross-fertilization of their respective strengths." (B) "we ... demonstrate how this connection facilitates the construction of streaming VMP." (C) "... shedding new light on the understanding and development of more advanced ABI techniques." The reviewer believes the main result alone is enough, and the paper would be better served by removing these claims. Additionally, (item 2) the claim that streaming VMP "is not straightforward and demands additional effort" is not justified; (item 4) the claim about insights into EP convergence should be removed or stated concretely; (item 5) online/distributed VMP is already well known.*
>
> **R1:** We appreciate the reviewer's careful reading and constructive feedback. To avoid potential ambiguity and ensure the claims precisely reflect the theoretical results, we have revised the relevant statements as follows:
>
> - **Claim A**: To avoid suggesting broader implications than established, this claim has been removed from the abstract and introduction. The revised text focuses on the concrete contributions: the algorithmic equivalence (Theorem 1) and the convergence guarantee for CEP (Corollary 1).
>
> - **Claim B / Item 2**: We have revised this claim to avoid the impression that we propose a new streaming VMP algorithm. Specifically, the sentence "is not straightforward and demands additional effort" has been removed. The relevant prior work on streaming variational Bayes is now properly included (Broderick et al. 2013; Masegosa et al. 2016). In the revised manuscript, we frame the streaming result as a natural consequence of the established equivalence: the equivalence provides a principled recipe by which any VMP-derived algorithm for model that satisfies the condition can inherit a streaming variant without additional derivation effort. This is demonstrated through the Bayesian tensor decomposition example in Section 4.4, where the streaming update drops out of the recipe and recovers the existing POST algorithm (Du et al. 2018) without re-derivation.
>
> - **Claim C**: This claim has been removed from the abstract, introduction, and conclusion to avoid suggesting implications beyond the scope of the current results.
>
> - **Item 4**: The claim regarding insights into the convergence of standard EP has been removed.
>
> - **Item 5**: The claim about online or distributed VMP has been removed. The relevant prior work is now properly cited.
>
> We believe these revisions ensure that the claimed contributions accurately reflect the scope of the theoretical results.
>
> *C2. (Item 1) Figure 1 is misleading, because it implies that ADF and the streaming version of CEP are the same thing. The equivalence is between streaming VI and streaming CEP, which is distinct from ADF. I think this needs to be corrected, e.g. by separately adding streaming CEP (or some other name), with a line showing that it is obtained from ADF using additional CEP approximations.*
>
> **R2:** We agree with the reviewer that the original Figure 1 conflated ADF with streaming CEP. In the revised manuscript, we have redrawn Figure 1 to include a separate *Streaming CEP* node, which is obtained from ADF by applying the surrogate distribution and delta approximation. The equivalence established in Corollary 2 is now correctly shown as between streaming CEP and streaming VMP, rather than between ADF and streaming VMP. Correspondingly, Corollary 2 has been restated to reflect this distinction, and the figure caption, text references, and the summary in Section 3.4 have all been updated to match.

---

> ### Author Response · Authors · 2026-04-15
>
> *C3. (Item 3) An equivalence between power EP and VI in the limiting case (of the power parameter) has been shown by Bui et al. 2018. This seems relevant and should be discussed somewhere.*
>
> **R3:** In the revised manuscript, we have added a discussion of Bui et al. (2017) in the related work paragraph of Section I, where Power EP and Rényi Divergence VI are discussed. We note that their result shows Power EP recovers VI in the limiting case of the power parameter, providing further evidence that VI and EP are closely related. We distinguish our contribution by noting that their result characterizes the limiting behavior of a generalized divergence, whereas our work establishes an explicit algorithmic equivalence between two distinct methods (VMP and CEP) under specific conditions.
>
> *C4. (Items 6--11) Presentation fixes: (6) missing summation in the $\\mathcal{L}(q)$ derivation; (7) "builds on ADF" -> "generalizes ADF"; (8) "corresponds to the likelihood" -> "may correspond to the likelihood"; (9) $\\tilde f\_i(\\boldsymbol{\\theta})$ and $\\tilde f\_i(\\boldsymbol{\\theta}\_m)$ are confusingly the same notation for different functions; (10) the statistics function $\\boldsymbol{\\phi}(\\boldsymbol{\\theta}\_m)$ is group-specific and should be subscripted; (11) "can be represented with a quadrature formula" -> "can be computed using quadrature methods."*
>
> **R4:** We thank the reviewer for these detailed suggestions. Below, we address each of the concerns:
>
> - **Item 6**: An intermediate step has been added to the $\\mathcal{L}(q)$ derivation, preserving the summation $\\sum\_j$ before it is absorbed into the constant.
> - **Item 7**: Changed to "generalizes ADF."
> - **Item 8**: Changed to "may correspond to the likelihood."
> - **Item 9**: We now use $\\tilde{f}\_{im}(\\boldsymbol{\\theta}\_m)$ consistently throughout the paper when referring to the group-specific subfactor, reserving $\\tilde{f}\_{i}(\\boldsymbol{\\theta})$ for the full (unfactorized) factor. A footnote has been added at the first definition to clarify this convention.
> - **Item 10**: Here we think that the notation $\\boldsymbol{\\phi}(\\boldsymbol{\\theta}\_m)$ without a group subscript would be better, as it is consistent with notation used in the original CEP paper. Since the group is always clear from the argument $\\boldsymbol{\\theta}\_m$, we believe this remains unambiguous.
> - **Item 11**: Changed to "can be computed using quadrature methods."
>
> *C5. (Item 12) "The conditional moment $\\mathbb{E}\_{\\hat{p}\_i(\\boldsymbol{\\theta}\_m|\\boldsymbol{\\theta}\_{\\setminus m})}[\\boldsymbol{\\phi}(\\boldsymbol{\\theta}\_m)]$ is a function of the sufficient statistics of $\\boldsymbol{\\theta}\_{\\setminus m}$." This is not necessarily true; it requires additional assumptions. Can you clarify what additional assumptions are being made here?*
>
> **R5:** We thank the reviewer for raising this point. The reviewer is correct that, in general, this property requires an additional assumption: namely, that the conditional density $\\hat{p}\_i(\\boldsymbol{\\theta}\_m | \\boldsymbol{\\theta}\_{\\setminus m})$ depends on $\\boldsymbol{\\theta}\_{\\setminus m}$ only through the sufficient statistics $\\boldsymbol{\\phi}(\\boldsymbol{\\theta}\_{\\setminus m})$. We acknowledge that this was not made explicit in the original text, as it was also left unspecified in the original CEP paper.
>
> In the revised manuscript, we have added a clarifying sentence at this location in Section II stating this condition explicitly. We further note that in conjugate-exponential models, which we assume in our main results (Section III), this property is automatically satisfied. Specifically, in such models, $\\ln p(\\boldsymbol{\\theta}\_m | \\boldsymbol{\\theta}\_{\\setminus m})$ is a multi-linear function of the sufficient statistics $\\boldsymbol{\\phi}(\\boldsymbol{\\theta}\_m)$ and $\\boldsymbol{\\phi}(\\boldsymbol{\\theta}\_{\\setminus m})$ (Winn & Bishop 2005), so the natural parameter $\\boldsymbol{\\eta}\_m$ depends on $\\boldsymbol{\\theta}\_{\\setminus m}$ only through $\\boldsymbol{\\phi}(\\boldsymbol{\\theta}\_{\\setminus m})$, and hence the conditional moment is indeed a function of $\\boldsymbol{\\phi}(\\boldsymbol{\\theta}\_{\\setminus m})$. This justification has also been added near the discussion of Theorem 1's conditions in Section III.

---

> ### Author Response · Authors · 2026-04-15
>
> *C6. (Item 13) $\\boldsymbol{\\phi}(\\cdot)$ is used to denote the statistic functions of both the approximating family and the conditional exponential family of the model. Are we assuming these functions are always the same?*
>
> **R6:** Yes, in the conjugate-exponential setting considered in this paper, the sufficient statistics of the variational distribution $q(\\boldsymbol{\\theta}\_m)$ and the conditional distribution $p(\\boldsymbol{\\theta}\_m | \\boldsymbol{\\theta}\_{\\setminus m}, \\mathcal{D})$ are identical. This is precisely the consequence of conjugacy: the posterior remains in the same exponential family as the prior, sharing the same sufficient statistics. Since this holds throughout our analysis, we use the same notation $\\boldsymbol{\\phi}(\\boldsymbol{\\theta}\_m)$ for both.
>
> *C7. (Items 14--16) (14) "where the tilted distribution $q^{\\setminus i}(\\boldsymbol{\\theta})$ is defined in (8)" should say cavity/calibrating distribution, and it is not defined in (8). (15) The notation $\\hat{p}\_i(\\boldsymbol{\\theta}\_m|\\mathbb{E}\_q[\\boldsymbol{\\phi}(\\boldsymbol{\\theta}\_{\\setminus m})])$ is not clear and should be defined explicitly before its first use. (16) "using $q(\\boldsymbol{\\theta}\_m)$ as a surrogate of $p(\\boldsymbol{\\theta}\_m)$" should be $q(\\boldsymbol{\\theta}\_{\\setminus m})$ and $p(\\boldsymbol{\\theta}\_{\\setminus m}|\\mathcal{D})$.*
>
> **R7:** All three have been corrected in the revised manuscript:
>
> - **Item 14**: "Tilted distribution" has been corrected to "calibrating distribution," and the incorrect equation reference has been replaced with a reference to the relevant definition in Section II.
> - **Item 15**: We have added an explicit definition of the notation $\\hat{p}\_i(\\boldsymbol{\\theta}\_m|\\mathbb{E}\_q[\\boldsymbol{\\phi}(\\boldsymbol{\\theta}\_{\\setminus m})])$ before its first use (prior to Lemma 4), clarifying that it denotes the conditional distribution $\\hat{p}\_i(\\boldsymbol{\\theta}\_m|\\boldsymbol{\\theta}\_{\\setminus m})$ with the sufficient statistics $\\boldsymbol{\\phi}(\\boldsymbol{\\theta}\_{\\setminus m})$ replaced by their expectations under $q$.
> - **Item 16**: Corrected to $q(\\boldsymbol{\\theta}\_{\\setminus m})$ and $p(\\boldsymbol{\\theta}\_{\\setminus m}|\\mathcal{D})$.
>
> *C8. (Item 17) The authors switch from using $\\mathbb{E}[\\cdot]$ to using $\\langle \\cdot \\rangle$ to denote expectations halfway through (25) and for the remainder of that section, for seemingly no reason. It gives the impression that separate pieces of work have been stitched together with no attempt to ensure consistency.*
>
> **R8:** We thank the reviewer for this suggestion. The use of $\\langle \\cdot \\rangle$ in Section IV is motivated by two considerations: first, it keeps the tensor decomposition formulas more concise and readable; second, it is consistent with the notation used in the existing literature on CEP and Bayesian tensor methods such as POST. In the revised manuscript, we have introduced this shorthand explicitly at the beginning of Section IV, before any equations, so that the notation is established upfront and the transition is clear.

---

> ### Author Response · Authors · 2026-04-15
>
> *C9. I have read the paper several times now and each read through I find multiple new errors. I would really recommend the authors carefully proofread the submission to check for errors.*
>
> **R9:** We thank the reviewer for the thorough reading. We have carefully proofread the entire manuscript and corrected the issues we identified, including typographical errors, grammatical issues, notation inconsistencies, and mathematical typos.
>
> *C10. I feel almost everything after page 12 is just filler content and does not really add anything to the paper -- I won't make drastic changes here a condition of my acceptance, but I would sincerely recommend the authors consider drastically cutting some of this content.*
>
> **R10:** We appreciate the reviewer's suggestion. We have taken concrete steps to condense the paper while retaining the content that we believe serves a useful role. Below, we first explain the motivation for keeping these sections, and then describe the specific reductions we have made.
>
> Section 3.3 (graphical model interpretation) provides an alternative view of our results, which is natural since VMP and EP/CEP are inherently tied to Bayesian networks and factor graphs, respectively. Section 3.4 (practical suggestions) offers guidance for practitioners on when and how to apply different ABI methods. The worked example (Section IV) provides a concrete demonstration of the equivalence and includes robustness experiments that Reviewer ogcC valued. Overall, the paper aims to be accessible to readers with different levels of familiarity with ABI methods.
>
> That said, following the reviewer's suggestion, we have substantially condensed the presentation. Specifically, the detailed graphical model derivation in Section 3.3 has been moved to a new appendix (Appendix D), with only the key results and final comparison retained in the main text. The summary paragraphs in Section 3.4 that recapped all lemmas and corollaries have been condensed to a single sentence referencing Figure 3. The intermediate CEP derivation steps in Section 4.3 (calibrating distribution, tilted distribution, conditional moments) have been moved to the tensor decomposition appendix (Appendix F.2), while the main text retains the setup, final results, and comparison with VMP. Additionally, several verbose paragraphs in Sections I--III have been tightened to reduce redundancy. We believe these changes make the paper more concise while remaining accessible to readers with different levels of background.

---

### Review · Reviewer_ogcC · 2026-04-08

**Summary Of Contributions:**

The paper’s main contribution is to identify a theoretical connection between two approximate Bayesian inference methods: variational message passing (VMP) and conditional expectation propagation (CEP), which are traditionally viewed as distinct.

The central result is that, under a specific set of assumptions, the two methods produce the same update equations. Building on this, the authors argue that CEP inherits a convergence guarantee in that setting and that the same connection helps derive a streaming form of VMP related to assumed density filtering (ADF). The authors also provide a graphical model interpretation of the result, using tensor decomposition as concrete examples.

Key Strengths
1 Provide a rigorous proof of equivalence between VMP and CEP, offering an elegant and unifying perspective on these two classic probabilistic message-passing algorithms.
2 Derive the convergence guarantee for CEP, which has traditionally been a major limitation for the applications of CEP.
3 The extension of the streaming VMP algorithm provides practically useful handling of sequential or large-scale data.

The main weakness is that the result depends on a fairly restrictive collection of assumptions, and the proof is under a much narrower scope than the title implies (general equivalence).
A minor weakness is that the empirical section is also limited, since it mainly validates the theory in a model class chosen to satisfy the assumptions used in proof, rather than testing the practical scope of the result more broadly.

**Audience:**

Yes

**Audience Explanation:**

Researchers working on approximate Bayesian inference, variational inference, expectation propagation, and graphical models would likely find the result useful, especially because it gives an algorithm-level connection between VMP and CEP.  The main result is probably specialized rather than broad, but it is clearly relevant to a meaningful part of the TMLR audience.

**Broader Impact Concerns:**

There are no significant broader impact concerns for this submission.

**Claims And Evidence:**

Yes

**Claims Explanation:**

Yes, the claims made in the submission are supported by accurate and convincing evidence.
The authors rigorously prove the claims using established mathematical lemmas regarding KL divergence and moment matching to prove the theoretical equivalence of VMP and CEP under specific conditions.

Furthermore, they provide clear empirical validation through a Bayesian tensor decomposition experiment for image completion

**Requested Changes:**

The title should be narrowed or made more precise, since the main equivalence and convergence results are proved only under a fairly restricted set of assumptions, especially the delta-based expectation approximation. In the current version,  the title suggests a broader connection or equivalence between VMP and CEP than the paper actually establishes. A revised title should better reflect that the contribution is a conditional equivalence result rather than a more general one.

Fix typos.

---

> ### Author Response · Authors · 2026-04-15
>
> We thank the reviewer for the thorough evaluation and for acknowledging the novelty of our main result. Below, we address each of the reviewer's comments.
>
> *C1. The title should be narrowed or made more precise, since the main equivalence and convergence results are proved only under a fairly restricted set of assumptions, especially the delta-based expectation approximation. In the current version, the title suggests a broader connection or equivalence between VMP and CEP than the paper actually establishes. A revised title should better reflect that the contribution is a conditional equivalence result rather than a more general one.*
>
> **R1:** We agree that the title can be narrowed to be more precise. In the revised manuscript, the title has been changed to *"When VMP Meets CEP: An Algorithmic Equivalence Under Mild Conditions"*, which explicitly conveys that the contribution is an algorithmic equivalence that holds under specific conditions. In addition, following the suggestions of Reviewer GneH, we have also revise the main context to more precisely reflect the scope of our contributions.
>
> *C2. Fix typos.*
>
> **R2:** We have carefully proofread the entire manuscript and corrected all typographical and grammatical errors. Additionally, we have standardized the notation and clarified certain terminology following the suggestions of Reviewer GneH.
>
> *C3. The empirical section is also limited, since it mainly validates the theory in a model class chosen to satisfy the assumptions used in proof, rather than testing the practical scope of the result more broadly.*
>
> **R3:** We appreciate this observation. While the primary purpose of the experiments is to validate the theoretical equivalence, we agree that investigating the robustness of the result beyond the stated assumptions is valuable. In fact, the current manuscript already includes two additional experiments designed for this purpose (see Figure 7 in the revised paper):
>
> - In the first experiment, we violate the group-wise update assumption by adopting an entry-wise update strategy for CEP. The results show that CEP still converges reliably, despite the lack of a formal convergence guarantee under this setting.
> - In the second experiment, we test performance under non-i.i.d. noise. Both VMP and CEP maintain identical RMSE levels, demonstrating stability beyond the i.i.d. assumption.
>
> Note that these experiments should be viewed as robustness checks rather than evidence for a broader theorem. Nevertheless, we believe they provide useful practical insights and help readers better understand the behavior of these ABI methods beyond the theoretical conditions.

---

### Decision · Action_Editor_ouB8 · 2026-05-11

**Recommendation:** Accept with minor revision

**Audience:**

Yes

**Audience Explanation:**

This paper should be interesting to probabilistic modelling community.

**Claims And Evidence:**

Yes

**Claims Explanation:**

This paper's contribution is mainly on the theoretical side: the authors have attempted to provide a connection between conditional expectation propagation (CEP) and variational message passing (VMP).

This re-submission has fixed the critical theoretical issue pointed out in last round of submissions. The AE recruited two of the previous reviewers to check the results, and they've confirmed that the issue has been fixed. They welcomed the theoretical contribution and stated that it would be of interest to TMLR community, especially those working on probabilistic inference methods.

The remaining concerns from the reviewers are still on the clarity of presentations, especially on precise grounding of the claims based on the theoretical results. Please consider reviewers' recommendations on e.g., revising some claims (to tone down the results) and/or removing some minor claims that are not well grounded by theory (or at least making them more precise).